# A Temporary Immersion System Improves Regeneration of In Vitro Irradiated Recalcitrant Indica Rice (*Oryza sativa* L.) Embryogenic Calli

**DOI:** 10.3390/plants11030375

**Published:** 2022-01-29

**Authors:** Alejandro Hernández-Soto, Jason Pérez, Rebeca Fait-Zúñiga, Randall Rojas-Vásquez, Andrés Gatica-Arias, Walter Vargas-Segura, Ana Abdelnour-Esquivel

**Affiliations:** 1Doctorado en Ciencias Naturales para el Desarrollo (DOCINADE), Instituto Tecnológico de Costa Rica, Universidad Nacional, Universidad Estatal a Distancia, Cartago P.O. Box 159-7050, Costa Rica; 2Biotechnology Research Center, Biology School, Costa Rica Institute of Technology, Cartago P.O. Box 159-7050, Costa Rica; jasperez@itcr.ac.cr (J.P.); rebe.fait@gmail.com (R.F.-Z.); aabdelnour@itcr.ac.cr (A.A.-E.); 3Plant Biotechnology Laboratory, School of Biology, University of Costa Rica, San José P.O. Box 2060, Costa Rica; randallrojas16@gmail.com (R.R.-V.); andres.gatica@ucr.ac.cr (A.G.-A.); 4Programa de Posgrado en Ciencias Agrícolas y Recursos Naturales (PPCARN), School of Agronomy, University of Costa Rica, San José P.O. Box 2060, Costa Rica; 5Vitroflora Labs S.A., Alajuela 20701, Costa Rica; 6Programa de Posgrado en Biología (PPB), School of Biology, University of Costa Rica, San José P.O. Box 2060, Costa Rica; 7Gamma Irradiation Laboratory, School of Physics, Costa Rica Institute of Technology, Cartago P.O. Box 159-7050, Costa Rica; walvargas@tec.ac.cr

**Keywords:** somatic embryogenesis, Cobalt-60, radiation-induced mutagenesis, temporary immersion systems (TIS)

## Abstract

The development of gamma ray-mutated rice lines is a solution for introducing genetic variability in indica rice varieties already being used by farmers. In vitro gamma ray (^60^Co) mutagenesis reduces chimeras and allows for a faster selection of desirable traits but requires the optimization of the laboratory procedure. The objectives of the present work were sequencing of *mat*K and *rbc*L, the in vitro establishment of recalcitrant rice embryogenic calli, the determination of their sensitivity to gamma radiation, and optimization of the generation procedure. All sequenced genes matched perfectly with previously reported *mat*K and *rbc*L *O. sativa* genes. Embryogenic calli induction improved using MS medium containing 2 mg L^−1^ 2,4-D, and regeneration was achieved with MS medium with 3 mg L^−1^ BA and 0.5 mg L^−1^ NAA. The optimized radiation condition was 60 Gy, (LD20 = 64 Gy) with 83% regeneration. An immersion system (RITA^®^, Saint-Mathieu-de-Tréviers, France) of either 60 or 120 s every 8 h allowed systematic and homogeneous total regeneration of the recalcitrant line. Other well-known recalcitrant cultivars, CR1821 and CR1113, also had improved regeneration in the immersion system. To our knowledge, this is the first study reporting the use of an immersion system to allow for the regeneration of gamma-ray mutants from recalcitrant indica rice materials.

## 1. Introduction

Rice is an important cereal that provides 20% of the world’s energy, particularly in Asia, Africa, and Latin America [1]. The *Oryza* genus consists of 22 species, but only two are commonly planted, namely, *O. sativa* and *O. glaberrima* [2,3,4,5,6]. Farmers prefer only a few cultivars, depending on the country. The limited genetic variability of commercial materials can become an obstacle in increasing productivity given emerging conditions such as heat, salt stress, soil acidification, plague sensitivity, and weeds. Introducing variability with crossbreeding is slow and can result in the introduction of undesired traits. Radiation methods to generate variability in seeds are standard techniques used since 1928 on vegetables, while other methods exist, such as ethyl methanesulfonate (EMS), and new breeding techniques to introduce specific genetically engineered mutations [7,8,9,10,11,12].

Plant-tissue culture represents an opportunity to overcome time limitations, land requirements, and the selection of the desired variability that results when using gamma-ray mutagenesis in seeds. Once irradiated, the seeds must be planted several times from M0 = seeds prior to mutagenic treatment; M1 = plants produced from material treated with the mutagen, to subsequent generations termed M2, M3, M4, to avoid chimeras and heterogenicity until exposed to stress-selection conditions, as has been done in the past [6,8]. In contrast, rice-tissue culture can produce a primitive cell aggregate or calli with embryogenic potential and consequently mutate and regenerate from one or a few cells with stressor selection from the beginning, such as NaCl or herbicide [11,13,14].

The latter is possible because plant cells are totipotent, which means that whole plants can develop from single cells [15]. We previously reported embryogenic calli mutation using gamma radiation for the Costa Rican cultivar CR-5272 for salt and drought tolerance; however, farmers no longer use this cultivar and instead use modern materials recalcitrant to tissue culture [14]. The establishment of embryogenic rice calli is influenced by the germplasm of origin and the 2,4-D concentration [15,16]. Embryogenic cells result when exposed to Murashige and Skoog (MS) medium supplemented with 2 to 2.5 mg L^−1^ 2,4-D, resulting in pro-embryos and somatic embryos [17,18,19]. Costa Rican cultivars such as CR-201, CR-1707, CR-1821, CR-8334, and CR-8341 have unpredictable and variable behavior, while CR-1113 and CR-5272 have predictable induction and regeneration in 2.5 mg L^−1^ 2,4-D [20].

Here, we faced three challenges. First, the identification of Lazarroz FL rice line with molecular markers, the chloroplast maturase gene K (*matK*) and ribulose-1,5-bisphosphate carboxylase small/large subunit (*rbcL*). Second, the improvement of our plant-tissue culture methods. Third, determining the radiosensitivity of embryogenic calli and further plant regeneration. We present a simple method to induce mutations using gamma rays in embryogenic calli of a recalcitrant cultivar, with an alternative immersion method that allowed our material to generate homogeneous and predictable in vitro plants after irradiation.

## 2. Results

### 2.1. Molecular Markers Used for the Identification of the Rice Cultivars

The chloroplast maturase gene K (*mat*K) and ribulose-1,5-bisphosphate carboxylase small/large subunit (*rbc*L), MZ558335 and MZ558334 sequences of Lazarroz FL showed perfect matches with the already published NCBI *Oryza sativa* indica demonstrated the identity of the non-irradiated material as expected. We detected three synonymous mutations in the *rbc*L sequence that are important for the characterization of the variety (Figure 1). Specifically, one synonymous SNP on the sixth glutamic acid triplet (GAA/GAG) indicates a putative origin of the germplasm ancestors from the Southeast Asia region because of its unique presence and matches with three cultivars of the region: the Pakistan cultivar NARC 17958 (GenBank KP827660.1), the Indonesia cultivar Pandak Kembang (GenBank MZ198248) and the Vietnam cultivar “Lua Khau Ky” isolate GBVN15800 (GenBank KR073275.1). We also detected two synonymous biallelic mutations at glycine 82 and 150 codifying triplets (GGC/GGA; GGT/GGC), which helped characterize and further identify the material. None of the mutations suggested biological importance since the open reading frame remained unaltered.

### 2.2. Embryogenic Calli Induction

The embryogenic calli induction in Lazarroz FL was affected by the combination of the plant growth regulators used. Thus, induction percentages ranging from 12.77 to 71.44 were obtained, with significant differences between all treatments. The higher calli induction percentage (71.44) was achieved with 2 mg L^−1^ 2,4-D alone (Table 1). In our case, the positive induction at 2 mg L^−1^ of 2,4-D contrasted with a higher brown callus rate (3.55%) and although it is a small value, for the next steps of regeneration represented a challenge and particularly in our final irradiation goal, which also triggers browning (Table 1).

Rice embryogenic calli obtained from induction medium supplemented with 2 mg L^−1^ were produced after 15 days of culture from the scutellum of mature zygotic embryos (Figure 2A) and were composed of yellow friable aggregates (Figure 2B).

### 2.3. Regeneration on Semisolid Medium

The best regeneration rate of approximately 70% resulted from 0.5 mg of NAA+ 3 mg of 6-BA, with sprouting of 7.14% and browning of only 9.52%, for calli induced on 2 mg L^−1^ 2,4-D (Table 2). Other regeneration medium recipes also resulted in regeneration but with a higher browning and lower sprouting rate, and were consequently useless for our next step, gamma radiation mutagenesis.

### 2.4. Gamma Radiation Mutagenesis

The effect of cobalt-60 (^60^Co) gamma radiation on embryogenic indica rice calli was evaluated. A lethal dose (LD50) of the embryogenic calli was found to be 110 Gy, while the 20% lethal dose was 64 Gy, resulting in a 0 to 120 gray gradient exposure, with 200 calli per exposure (Table 3).

The best radiation/regeneration ratio was achieved at 60 Gy with 83% regeneration, allowing for a balance between gamma radiation at lethal dose 20 and regeneration (Figure 3). The increased regeneration achieved with 40 Gy (75.00%) and 60 Gy (83.85%) versus the control (69.04%) (Figure 3A) is a hormetic behavior previously reported in our lab [14].

Compared to the non-irradiated control (0 Gy), a significant decrease in the sprouting of embryogenic calli was observed, as the dose increased from 0 to 80 Gy. Sprouting of embryogenic calli irradiated with gamma rays (^60^Co) decreased significantly (*p* < 0.05) occurred at doses higher than 60 Gy (Figure 3B). Moreover, Gamma-ray doses higher than 60 Gy severely affected browning rate (Figure 3C).

Gamma-irradiated calli produced plants after 45 days in the regeneration medium (Figure 4A) and fully in vitro plants (Figure 4B) at 60 days post-irradiation at 60 Gy.

### 2.5. Regeneration in Recipient for Automated Temporary Immersion (RITA^®^ Saint-Mathieu-de-Tréviers, France)

Regeneration in the MS semisolid medium was irregular and not homogenous (Figure 5A). Instead, the RITA^®^ immersion system (RITA^®^, Saint-Mathieu-de-Tréviers, France) provided predictable and homogeneous regeneration (Figure 5B).

The potential for temporary immersion system to regenerate recalcitrant materials (CR-5272, CR-1821, CR-1113 and Lazarroz FL) was evaluated. Our results showed no significant differences between the immersion time (60 vs. 120 s) on the regeneration capacity, sprouting and browning (Table 4). Nevertheless, a higher regeneration rate was obtained using the RITA^®^ (Saint-Mathieu-de-Tréviers, France) in 60 s (100%) compared to regeneration on the best semi-solid medium (70%). Similarly, the sprouting rate was higher using the RITA^®^ system independently of the immersion time compared to that obtained using the selected semi-solid medium (Table 2 and Table 4).

Although, the immersion time (60 vs. 120 s) did not significantly affect the browning rate, and it was higher using an immersion of 120 s (97.56%) compared to 60 s (60.00%) and a semi-solid medium (9.52%) (Table 2 and Table 4).

The potential for a temporary immersion system to regenerate recalcitrant materials was also validated for the well-known recalcitrant cultivars CR-5272, CR-1821 and CR-1113 (Figure 6).

## 3. Discussion

The present work optimized in vitro gamma ray (^60^Co) mutagenesis in an embryogenic calli of a recalcitrant Costa Rican Lazarroz FL cultivar. The cultivar seems to be related to Southeast Asian rice cultivars based on its *rbc*L sequence pattern.

The best calli induction was observed for MS with 2mg of 2,4-D and regeneration with MS with 0.5 mg ANA + 3 mg BA. The calli induction step with a 2 mg L^−1^ of 2,4-D concentration was initially not expected because of previous local cultivar reports. Local cultivars CR-5272 and CR-1113 had a positive response at 2.5 mg L^−1^ of 2.4 D but a low performance at 2 mg L^−1^ of 2,4-D, while CR-201, CR-1707, CR-1821, CR-8334, and CR-8341 had recalcitrant and unpredictable in vitro behaviors [17,20]. Our result is similar to that obtained by other authors on Southeast Asian cultivars such as Malaysia MR219, where 2,4-D is critical and performs the best as an inducer at 2 mg L^−1^ [21,22]. On the contrary, a better calli induction occurred with higher concentrations of 2,4-D (from 2.5 mg to 3 mg L^−1^) with other cultivars, such as MR220, GNY-53, and JP-5 [23,24].

Regeneration was improved using a temporal immersion system (RITA^®^, Saint-Mathieu-de-Tréviers, France) with either 60 or 120 s of immersion every 8 h, which achieved a predictable and more homogeneous regeneration response. The radiation dose with which to start mutagenesis was proposed as a lethal dose 20 at 60 Gy, as regeneration was not affected but remained as high as 80%. At 80 Gy, the regeneration fell to 30%, consistent with the oxidative damage provoked by radiation and corresponding to the lethal dose 30 of the embryogenic calli. We believe 60 Gy is an excellent condition to start mutagenesis, considering that in other in vitro plants, such as pineapple, potato, and banana, a 5–40 Gy dose of gamma irradiation was shown to be sufficient to produce variability [25]. The calculated radiation dose of the Lazarroz FL cultivar at 60 Gy corresponded to the LD20 and was different from our previous data achieved for CR-5272 with an LD50 of 60Gy [14].

Optimization of embryogenic calli in our recalcitrant Lazarroz FL cultivar became a challenge while at the same time an opportunity compared to our previous results from CR-5272 for the following reasons First, the alternative immersion method allowed our material to generate homogeneous and predictable in vitro plants after irradiation. The technique enabled regeneration for our Lazarroz FL cultivar and other recalcitrant rice lines CR-1821 and CR-1113, and consequently, is a great potential tool for regenerating gamma radiation mutated materials. The constant liquid and airflow can dilute oxidative compounds and facilitate more homogeneous exposure to nutrients. We validated the results with recalcitrant cultivars CR-1821 and CR-1113, which showed a total regeneration rate in contrast to the null regeneration in the conventional semisolid method. Plant cultivars’ nonhomogeneous in vitro behavior is not fully understood, but recent discoveries provide insights into the genetic basis of the sucrose metabolism and calli browning. External phytohormones seem to trigger rice sucrose metabolism required for regeneration. The system appears to rely on the expression of endogenous cytokinin, auxin, and ABA signaling genes: ORYZA SATIVA RESPONSE REGULATOR 1 (ORR1), PIN-formed 1 (PIN1), and *late embryogenesis-abundant 1*(LEA-1) [26]. The expression of OsSRO1c, a regulator of oxidative stress, seems to be vital for avoiding calli browning in indica cultivars [27]. Tissue culture and gamma radiation produce oxidation via reactive oxygen species (ROS), usually contained in chloroplasts, peroxisomes, and mitochondria. ROS include superoxide (O_2_^−^), hydroxyl (OH^−^) radicals, and hydrogen peroxide (H_2_O_2_) [28]. Tissue culture and gamma rays trigger ROS, consequently damaging the DNA by oxidation of the molecule into 8-oxo-7-hydroxyguanosine (8-oxo-dG) and further transversions of C/G and T/A [29].

Second, we can develop the desired mutations in a cultivar that farmers are already using, leading to faster breeding and adoption of a derived improved cultivar. Rice traits associated with specific genes are well known, which paves the way for producing novel cultivars based on mutation of desired or required conditions, such as biotic and abiotic stress tolerance [11]. We foresee the development of new traits such as NaCl, herbicide, and pH tolerance based on the corresponding selection agents and not limited by recalcitrant cultivars, which used to be our bottleneck to bring innovation.

## 4. Materials and Methods

### 4.1. Molecular Markers

A NucleoSpin^TM^ Tissue Kit Macherey-Nagel (Düren, Germany) was used for DNA extraction from 1 mg of on non-irradiated lyophilized leaf tissue of Lazarroz FL. Thermo Fisher K1071 (Vilnius, Lithuania) was used for the subsequent PCR following the recommendations of the manufacturer. The primers used in this study are as follows: for rbcL, rbcLaf 5′ATGTCACCACAAACAGAGACTAAAGC3′ and rbcLar 5′GTAAAATCAAGTCCACCRCG-3′ or rbcLaf 5′ATGTCACCACAAACAGAGACTAAAGC3′ and rbcLr590 5′AGTCCACCGCGTAGACATTCAT-3′; for matK, matK-xf 5′TAATTTACGATCAATTCATTC-3′ and matKr 5′ACAAGAAAGTCGAAGTAT-3′. Briefly, the PCR master mix consisted of a mixture (50 μL) containing 1X Dream Taq Master mix Thermo Fisher (Vilnius, Lithuania), 20 μM of each primer, and 5 μL of DNA (50 ng/uL). The thermocycling program was 95 °C for 5 min, 40 cycles at 95 °C for 45 s, 55 °C for 45 s and 72 °C for 1 min, and a final cycle of 72 °C for 7 min.

### 4.2. Embryogenic Calli Induction

For initial assays, the palea and lemma of rice caryopsis of a commercial local indica cultivar were removed with No. 80 grit sandpaper. The obtained seeds were surface-sterilized as previously reported [14] through two incubations in 4% (*v*/*v*) NaOCl for 10 min each with constant agitation, using 10 mL of disinfectant solution for every gram of seeds. After the first and final incubation, the seeds were washed six to seven times with distilled sterilized water and were cultured in media composed of mineral salts and vitamins as described by Murashige and Skoog (MS), with 20 g L^−1^ sucrose and 0.1 g L^−1^ hydrolyzed casein. Treatments consisted of supplementing the basal medium with one of the following combinations of plant growth regulators: (i) 2.5 mg L^−1^ 2,4-dichlorophenoxyacetic acid (2,4-D) was used as a control, (ii) 2.0 mg L^−1^ 2,4-D, (iii) 1.0 mg L^−1^ 2,4-D + 1.0 mg L^−1^ 6- benzyladenine (6-BA), (iv) 2.0 mg L^−1^ 2,4-D + 1 mg L^−1^ 6-BA, and (v) 2.0 mg L^−1^ 2,4-D + 0.25 mg L^−1^ 1-phenyl-3-(1,2,3-thidiazol-5-yl)urea (thidiazuron, TDZ) (Table 1). After adding plant growth regulators, the media volumes were adjusted as required, the pH was adjusted to 5.8 with 1 N NaOH or 1 N HCl, and 5.4 g L^−1^ Gelzan^®^ (Phytotechnology Laboratories^®^, Shawnee Mission, KS, USA) were added as a gelling agent. All previously mentioned chemicals were supplied by Phytotechnology Laboratories^®^ (Shawnee Mission, KS, USA). An autoclave (1.2 ATM. cm^−2^ and 121 °C for 30 min) was used for the sterilizing medium and further dispensed on 94 × 16 mm vented polystyrene Petri dishes (Greiner Bio-One, Fisher-Scientific, Waltham, MA, USA) in a laminar flow chamber. For each treatment, at least 900 seeds were cultured after surface sterilization. Cultures were maintained in the dark at 26 ± 2 °C. Calli induction and browning rates (brown or necrotic/total calli) were recorded as response variables and analyzed in a completely randomized design with a generalized linear model with a Poisson distribution and logit link function. Post hoc analysis consisted of an Honest Significant Difference test on IBM SPSS version 27 [30] and differences between each combination of factors were recognized.

### 4.3. Regeneration on Semisolid Medium

Embryogenic calli obtained from each induction medium were transferred to different regeneration treatments. Basal regeneration medium was similar to that described by Sudhakar et al. [31], and was constituted by MS mineral salts and vitamins, 20 g L^−^^1^ sucrose and 0.3 g L^−^^1^ hydrolyzed casein. Treatments consisting of supplementing the basal medium with variations of growth regulators, and the control was supplemented with 0.5 mg L^−^^1^ α-Naphthaleneacetic Acid (NAA) + 3 mg L^−^^1^ 6-BA, while the second treatment was supplemented with 0.5 mg L^−^^1^ NAA + 0.5 mg L^−^^1^ 6-BA and the third treatment was supplemented with 0.5 mg L^−^^1^ NAA + 1.5 mg L^−^^1^ Kinetin (KIN) and a fourth treatment consisted of supplementing basal medium with 0.5 mg L^−^^1^ NAA + 0.5 mg L^−^^1^ TDZ. After adding growth regulators, pH was adjusted to 5.8 with 1 N NaOH or 1 N HCl and 5.4 g L^−^^1^ Gelzan ^®^ (Phytotechnology Laboratories^®^, Shawnee Mission, KS, USA) were added as gelling agent. All previously mentioned reagents were supplied by Phytotechnology Laboratories^®^ (Shawnee Mission, KS, USA). After a properly dissolving of gelling agent, 60 mL of media were dispensed on 475 mL polypropylene WNA Deli Containers, and afterwards, media were sterilized at 1.2 ATM.cm^−^^2^ and 121 °C for 30 min. For each treatment, 6 replicates of 7 calli were cultured on a factorial design. Factor 1 consisted of induction media and factor 2 was regeneration media. Cultures were maintained at an irradiance of 72 µmol s^−^^1^m^−^^2^, a 16 h light/8 h dark photoperiod was used and 26 ± 2 °C for a period of 4 weeks. Calli showing mature coleoptilar germinated embryos were determined as regenerated, calli with completely differentiated plantlets greater than 1 cm were categorized as sprouted, and necrotic calli with a dark coloration were identified as browning calli. These response variables were analyzed with a generalized linear model with a Poisson distribution and logit link function. Post hoc analysis consisted of an Honest Significant Difference test on IBM SPSS version 27 [31] and differences between each combination of factors were recognized.

### 4.4. Gamma Irradiation

Embryogenic calli irradiation was achieved with a gamma irradiator Ob-Servo Ignis type with 24 cobalt 60 source pencils (Institute of Isotopes Co, Ltd., Budapest, Hungary). To determine radiosensitivity and the median lethal dose (LD50), calli were irradiated at 0, 40, 60, 80, 100, 120 Gy. Ten repetitions, and 20 embryogenic calli per exposure were used. The survival rate was recorded after calli were transferred to the regeneration medium composed by basal regeneration medium supplemented with 0.5 mg L^−1^ NAA *+* 3 mg L^−1^ 6-BA, selected after previous experiments were analyzed. Culture conditions were the same indicated above and after 4 weeks lethal doses were calculated using probit analysis on IBM SPSS version 27 [31] based on calli death rates.

### 4.5. Regeneration in Recipient for Automated Temporary Immersion (RITA^®^, Saint-Mathieu-de-Tréviers, France)

The RITA^®^ (Saint-Mathieu-de-Tréviers, France) temporary immersion system regeneration of the embryogenic calli consisted of 200 mL regeneration media previously described in 4.3, with 23 four-week-old calli per unit, different rice cultivars (Lazarroz FL, CR-5272, CR-1821, and CR-1113). Light and temperature culture conditions remained unchanged, with 60 or 120 s immersion used as treatments, every eight hours. After four weeks of culture, the variables evaluated were regeneration, sprouting, and browning, green area calculations with ImageJ version 1.52p [32].

## 5. Conclusions

Rice tissue culture is a tool for conventional and modern breeding, but is limited to the genotype response, particularly during the regeneration of recalcitrant varieties. Our results collected using an immersion system helped to overcome such difficulties and allowed for the induction of gamma-ray mutants. A temporary immersion system seems to help overcome calli browning while allowing the tissue to recover and consequently presenting a more efficient method. We foresee that having access to such methods could diminish the time to trigger innovation and focus on selecting mutants with desired traits in commercially used varieties.

## Figures and Tables

**Figure 1 plants-11-00375-f001:**
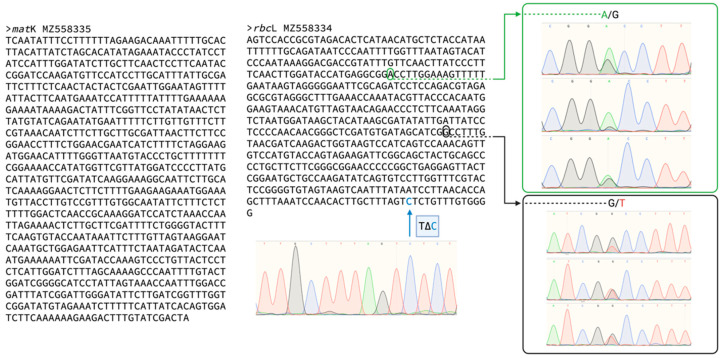
DNA markers used to identify the *mat*K and *rbc*L genes in non-irradiated Lazarroz FL rice variety. Note in green the synonymous SNP (C/T) and the biallelic synonymous mutations A/G and G/T (circled). None of the mutations had biological importance but helped in the genetic characterization of the cultivars.

**Figure 2 plants-11-00375-f002:**
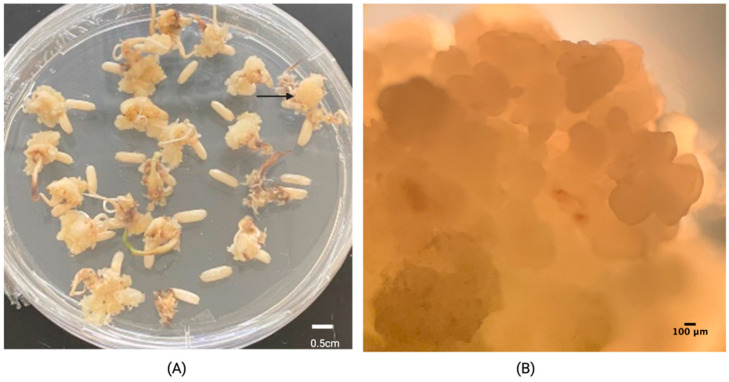
Calli induction of Lazarroz FL cultivar on MS medium with 2 mg L^−1^ 2,4-D after 15 days of culture under darkness. (**A**) Embryogenic calli obtained from the scutellum of mature zygotic embryos (arrow) (**B**) the compact and friable calli as observed under the stereoscope.

**Figure 3 plants-11-00375-f003:**
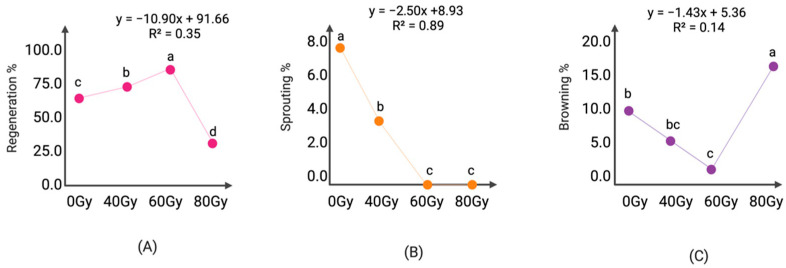
Correlation comparison of radiation dose influence on regeneration (**A**), sprouting (**B**) and browning rates (**C**) of calli of Lazarroz Fl cultivar, after 30 days of culture on regeneration medium. Letters a, b, bc, c and d represent significant differences for Tukey’s test (*p* ≤ 0.05). All treatments had 12 replicates of 7 calli each replicate (*n* = 84). Data were compiled at 15 days post-radiation.

**Figure 4 plants-11-00375-f004:**
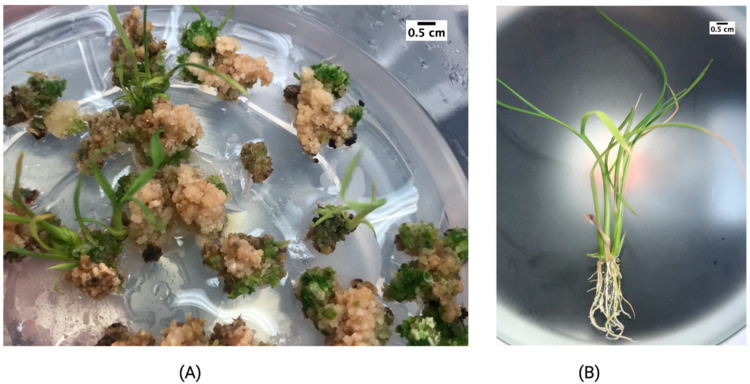
Response of 60 Gy irradiated calli Lazarroz FL cultivar after (**A**) 45 days and (**B**) 60 days of culture on regeneration medium.

**Figure 5 plants-11-00375-f005:**
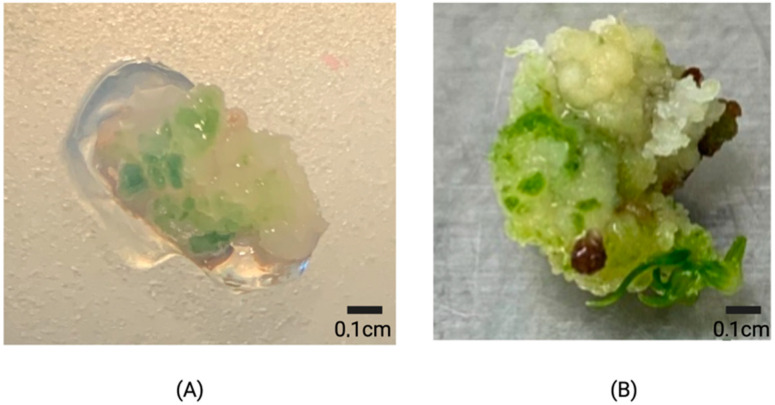
Regeneration in MS medium with 0.5 mg L^−1^ of NAA + 3 mg L^−1^ BA of an induced calli with 2 mg L^−1^ of 2,4-D after 15 days of induction. Regeneration in (**A**) semisolid medium and (**B**) RITA^®^ (Saint-Mathieu-de-Tréviers, France).

**Figure 6 plants-11-00375-f006:**
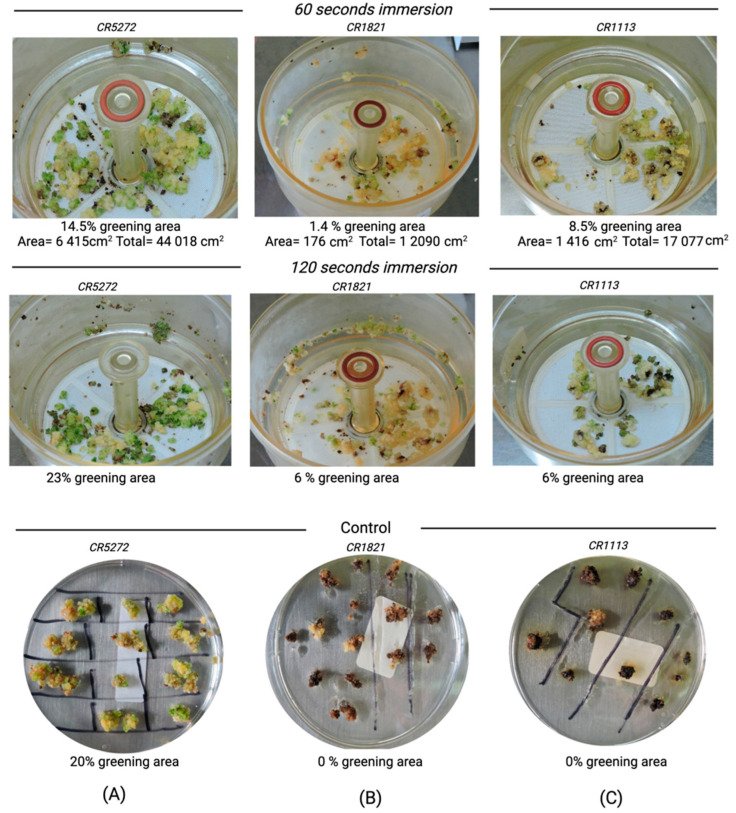
Regeneration of cultivars CR-5272 and recalcitrant CR-1821 and CR-1113 in MS medium with 0.5 mg L^−1^ of NAA + 3 mg L^−1^ of BA from calli induced with 2 mg L^−1^ of 2,4-D. (**A**) CR-5272 regeneration in RITA with immersion for 30 s or 60 s, and the semisolid medium control. (**B**) CR-1821 regeneration in RITA with immersion for 30 s or 60 s and the semisolid medium control. (**C**) CR-1113 regeneration in RITA with immersion for 30 s or 60 s and the semisolid medium control. Note that all RITA treatments contain calli with green areas that regenerate into plants. The absence of regeneration of the recalcitrant cultivars CR-1821 and CR-1113 was observed in the semisolid media.

**Table 1 plants-11-00375-t001:** Rice calli induction and browning rate from different induction treatments.

Treatment ^1^	id	Embryogenic Calli (%)	Browning Rate (%)
2.5 mg L^−1^ 2,4-D	i	21.44 b	2.00 b
2.0 mg L^−1^ 2,4-D	ii	71.44 a	3.55 a
1.0 mg L^−1^ 2,4-D + 1.0 mg L^−1^ BA	iii	12.77 d	0.66 c
2.0 mg L^−1^ 2,4-D + 1.0 mg L^−1^ BA	iv	16.77 c	0.21 c
2.0 mg L^−1^ 2,4-D + 0.25 mg L^−1^ TDZ	v	23.00 b	0.66 c

^1^ All treatments had 30 replicates of 30 seeds per replicate (*n* = 900). Letters represent a significant difference (*p* ≤ 0.05).

**Table 2 plants-11-00375-t002:** Rice calli response of Lazarroz FL cultivar after 4 weeks of culture on different regeneration media.

Induction Treatment ^1^	Regeneration Treatment	Regeneration (%)	Sprouting (%)	Browning (%)
2 mg 2,4-D	0.5 mg NAA + 3 mg BA	69.04 a	7.14 ab	9.52 d
	0.5 mg NAA + 0.5 mg TDZ	38.09 c	2.38 b	61.90 a
	0.5 mg NAA + 0.5 mg Kinetin	47.61 b	9.52 a	23.80 c
	0.5 mg NAA + 0.5 mg BA	28.57 d	2.38 b	54.76 b
1 mg BA + 2 mg 2,4-D	0.5 mg NAA + 3 mg BA	28.29 b	0 a	58.43 b
	0.5 mg NAA + 0.25 mg TDZ	58.82 a	0 a	100 a
	0.5 mg NAA + 0.5 mg Kinetin	9.22 c	0 a	56.81 b
	0.5 mg NAA + 0.5 mg BA	12.82 c	0 a	44.26 c
1 mg BA + 1 mg 2,4-D	0.5 mg de NAA + 3 mg BA	58.45 ab	16.38 a	18. 69 bc
	0.5 mg de NAA + 0.5 mg TDZ	61.75 a	10.71 ab	27.93 a
	0.5 mg de NAA + 0.5 mg Kinetin	56.31 ab	15.92 a	20.01 ab
	0.5 mg de NAA + 0.5 mg BA	49.88 b	4.16 b	11.66 c
	0.5 mg de NAA + 3 mg BA	58.45 ab	16.38 a	18. 69 bc
2.5 mg 2,4-D	0.5 mg NAA + 3 mg BA	34.64 b	0 b	9.20 c
	0.5 mg NAA + 0.5 mg TDZ	51.41 a	9.61 a	22.96 a
	0.5 mg NAA + 0.5 mg Kinetin	43.62 ab	0 b	15.73 b
	0.5 mg NAA + 0.5 mg BA	18.00 c	0 b	15.19 b
2 mg 2,4-D + 0.25 mg TDZ	0.5 mg NAA + 3 mg BA	77.27 a	2.27 a	96.59 a
	0.5 mg NAA + 0.5 mg TDZ	50.25 b	0 a	86.36 b
	0.5 mg NAA + 0.5 mg Kinetin	73.86 a	3.40 a	82.95 b
	0.5 mg NAA + 0.5 mg BA	44.29 b	0 a	72.81 c

^1^ All treatments had 6 replicates of 7 calli each replicate (*n* = 42). Letters represent significant differences for Tukey’s test (*p* ≤ 0.05) and the comparisons were made between treatments from same induction medium.

**Table 3 plants-11-00375-t003:** Lethal effect of gamma radiation on the embryogenic rice calli of Lazarroz FL cultivar after 30 days of culture on regeneration medium, determined by probit model of survival (%) ^1^.

Lethal Gamma Rays Dose Model	Dose (Gy)	Lower Limit (Gy)	Upper Limit (Gy)
LD10	41.145	34.552	46.708
LD20	64.799	60.083	69.20
LD25	73.785	69.388	78.139
LD30	81.855	77.507	86.403
LD40	96.429	91.649	101.85
LD50	110.050	104.435	116.720

^1^ All treatments had 10 replicates with 20 calli each (*n* = 200), *p* ≤ 0.05. Data were compiled at 30 days post-radiation.

**Table 4 plants-11-00375-t004:** Immersion regeneration, sprouting and browning rates ^1^.

Immersion Time	Regeneration	Sprouting	Browning Rate
60 s	100.00 a	25.00 a	60.00 a
120 s	97.56 a	31.71 a	97.56 a

^1^ All treatments had 4 replicates of 10 calli per replicate (*n* = 40), *p* ≤ 0.05. Data were compiled at 15 days post-radiation.

## Data Availability

Data is contained within the article.

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
