# Peer review of "A Temporary Immersion System Improves Regeneration of In Vitro Irradiated Recalcitrant Indica Rice (Oryza sativa L.) Embryogenic Calli"

_plants, 2022, doi:10.3390/plants11030375_

Round 1

Reviewer 1 Report

The manuscript is not comprehensive and difficult to read, particularly because of the lack of detailed explanation in Results and M&M.  More thorough explanation is needed in M&M and/or Results, for example, regeneration (%), sprouting(%), oxidation (%), RITA and SIT etc. 

A major concern is the lack of mutation-inducing effects of gamma-ray irradiation.  MATK and Rubisco sequences were examined and several SNPs were detected but they were all already known and used as local cultivar-specific markers.  

Explanation is needed on what were new or additional findings obtained in this experiment as compared with the ones reported in the previous paper Plant 2020, 56 (1), 88–97.  

Author Response

Thank you very much for all the inputs.
We have improved the document in content and format.

Your specific suggestions were incorporated as follows.

  1. The manuscript is not comprehensive and difficult to read, particularly because of the lack of detailed explanation in Results and M&M.  More thorough explanation is needed in M&M and/or Results, for example, regeneration (%), sprouting(%), oxidation (%), RITA and SIT etc. 

Accepted and corrected. The document was adjusted as requested, particularly MM.

Materials and Methods

4.1. Molecular markers

A NucleoSpinTM Tissue Kit (Macherey-Nagel, Germany) was used for DNA extraction from 1 mg of on non-irradiated lyophilized leaf tissue of Lazarroz FL. Thermo Fisher K1071 was used for the subsequent PCR following the recommendations of the manufacturer. The primers used in this study are described as follows: for rbcL, rbcLaf 5’ATGTCACCACAAACAGAGACTAAAGC3’ and rbcLar 5’GTAAAATCAAGTCCACCRCG-3’ or rbcLaf 5’ATGTCACCACAAACAGAGACTAAAGC3’ and rbcLr590 5’AGTCCACCGCGTAGACATTCAT-3'; for matK, matK-xf 5’TAATTTACGATCAATTCATTC-3’ and matKr 5’ACAAGAAAGTCGAAGTAT-3’. Briefly, the PCR master mix consisted of a mixture (50 μl) containing 1X Dream Taq Master mix (Thermo Fisher, 20 μM of each primer, and 5 μL of DNA (50 ng/uL). The thermocycling program was 95 °C for 5 min, 40 cycles at 95 °C for 45 s, 55 °C for 45 s and 72 °C for 1 min, and a final cycle of 72 °C for 7 min.

4.2. Embryogenic callus induction

For initial assays, the palea and lemma of rice caryopsis of a commercial local indica cultivar were removed with No. 80 grit sandpaper. The obtained seeds were surface-sterilized as previously reported13 through two incubations in 4% (v/v) NaOCl for 10 min each with constant agitation, using 10 mL of disinfectant solution for every gram of seeds. After the first and final incubation, the seeds were washed six to seven times with distilled sterilized water and were cultured in media composed of mineral salts and vitamins as described by Murashige and Skoog (MS), with 20 g L-1 sucrose and 0.1 g L-1 hydrolyzed casein. Treatments consisted of supplementing the basal medium with one of the following combinations of plant growth regulators: : (i) 2.5 mg L-1 2,4-dichlorophenoxyacetic acid (2,4-D) was used as a control, (ii) 2.0 mg L-1 2,4-D, (iii) 1.0 mg L-1 2,4-D + 1 mg L-1 6- benzyladenine (6-BA), (iv) 2.0 mg L-1 2,4-D + 1 mg. L-1 6-BA, and (v) 2.0 mg L-1 2,4-D + 0.25 mg L-1 1-phenyl-3-(1,2,3-thidiazol-5-yl)urea (thidiazuron, TDZ) (Table 1).. After adding plant growth regulators, the media volumes were adjusted as required, the pH was adjusted to 5.8 with 1 N NaOH or 1 N HCl, and 5.4 g L-1 Gelzan ® were added as a gelling agent. All previously mentioned chemicals were supplied by Phytotechnology Laboratories® (Shawnee Mission, Kansas, USA). An autoclave (1.2 ATM.cm-2 and 121 °C for 30 min) was used for sterilizing medium and further dispensed on 94x16 mm vented polystyrene Petri dishes (Greiner Bio-One, Fisher-Scientific, Waltham, Massachusetts, USA) in a laminar flow chamber. For each treatment, at least 900 seeds were cultured after surface sterilization. Cultures were maintained at darkness a at 26 ± 2 °C. Callus induction, oxidation rates (brown or necrotic/total calli) were recorded as response variables and analyzed in a completely randomized design with a generalized linear model with a Poisson distribution and logit link function. Post hoc analysis consisted of an Honest Significant Difference test to establish differences between the means.

4.3 Regeneration on semisolid medium

Embryogenic calli obtained from each induction medium were transferred to different regeneration treatments. Basal regeneration medium was similar to that described by Sudhakar et al. [30], and was constituted by MS mineral salts and vitamins, 20 g.L-1 sucrose and 0.3 g.L-1 hydrolyzed casein. Treatments consisted on supplementing the basal medium with variations on growth regulators: control was supplemented with 0.5 mg L-1 α-Naphthaleneacetic Acid (NAA) + 3 mg L-1 6-BA, second treatment was supplemented with 0.5 mg L-1 NAA + 0.5mg.L-1 6-BA, third treatment was supplemented with 0.5 mg L-1 NAA + 1.5 mg L-1 Kinetin (KIN) and a fourth treatment consisted on supplementing basal medium with 0.5 mg L-1 NAA + 0.5 mg L-1 TDZ. After adding growth regulators, pH was adjusted to 5.8 with 1 N NaOH or 1 N HCl and 5.4 g L-1 Gelzan ® were added as gelling agent. All previously mentioned reagents were supplied by Phytotechnology Laboratories. After a properly dissolving of gelling agent, 60 mL of media were dispensed on 475 mL polypropylene WNA Deli Containers, and afterwards, media were sterilized at 1.2 ATM.cm-2 and 121 C for 30 min. For each treatment, 6 replicates of 7 calli were cultured on a factorial design. Factor 1 consisted of induction media and factor 2 was regeneration media. Cultures were maintained at an irradiance of 72 µmol s-1m-2, a 16hrs light/8 hrs dark photoperiod was used and 26 ± 2 °C for a period of 4 weeks. Calli showing mature coleoptilar germinated embryos were determined as regenerated, calli with completely differentiated plantlets greater than 1 cm were categorized as sprouted, and necrotic calli with dark coloration were identified as oxidated calli. These response variables were analyzed with a generalized linear model with a Poisson distribution and logit link function. Post hoc analysis consisted of an Honest Significant Difference test and differences between each combination of factors were recognized.

4.4. Gamma irradiation

Embryogenic calli irradiation was achieved with a CoS 44HH-N Ob-Servo Ignis with 24 cobalt 60 sources (Institute of Isotopes Co, Ltd., Budapest, Hungary).To determine radiosensitivity and the median lethal dose (LD50), calli were irradiated at 0, 40, 60, 80 Gy. Ten repetitions, and 20 embryogenic calli per exposure were used. Survival rate was recorded after calli were transferred to regeneration medium composed by basal regeneration medium supplemented with 0.5 mg L-1 NAA + 3 mg L-1 6-BA, selected after previous experiments were analyzed. Culture conditions were the same indicated above and after 4 weeks lethal doses were calculated using probit analysis on IBM SPSS version 27 based on callus death rates.

4.5. Regeneration in Recipient for Automated Temporary Immersion (RITA®)

The RITA® temporary immersion system regeneration of the embryogenic calli consisted of 200 mL regeneration media previously described in 4.3, with 23 four-week-old calli per unit, different rice cultivars (Lazarroz FL, CR-5272, CR-1821, and CR-1113). Light and temperature culture conditions remained unchanged, with 60 or 120 seconds immersion as treatments, every eight hours. After four weeks of culture, the variables evaluated were regeneration, sprouting, and oxidation, green area calculations with ImageJ version 1.52p.

  1. A major concern is the lack of mutation-inducing effects of gamma-ray irradiation.  MATK and Rubisco sequences were examined and several SNPs were detected but they were all already known and used as local cultivar-specific markers.  

Accepted and corrected. The data correspond to non-irradiated material.

The chloroplast maturase gene K (matK) and ribulose-1,5-bisphosphate carboxylase small/large subunit (rbcL), MZ558335 and MZ558334 sequences of Lazarroz FL showed perfect matches with the already published NCBI O. sativa indica accessions and demonstrated the identity of the non-irradiated material as expected. We detected three synonymous mutations in the rbcL sequence that are important for the characterization of the variety (Fig 1).

  1. Explanation is needed on what were new or additional findings obtained in this experiment as compared with the ones reported in the previous paper Plant 2020, 56 (1), 88–97.  

Accepted and corrected.

Regeneration was improved using a temporal immersion system (RITA®) with either 60 or 120 seconds of immersion every 8 hours, which achieved a predictable and more homogeneous regeneration response. The radiation dose at which to start mutagenesis was proposed as a lethal dose 20 at 60 Gy, as regeneration was not affected but remained as high as 80%. At 80 Gy, the regeneration fell to 30%, consistent with the oxidative damage provoked by radiation and corresponding to the lethal dose 30 of the embryogenic calli. We believe 60 Gy is an excellent condition to start mutagenesis, considering that in other in vitro plants, such as pineapple, potato, and banana, a 5-40 Gy dose of gamma irradiation was shown to be sufficient to produce variability [23]. The calculated radiation dose of the Lazarroz FL cultivar at 60 Gy correspond to the LD20 and was different from our previous data achieved for CR-5272 having a LD50 of 60Gy[14].

Optimization of embryogenic calli of our recalcitrant Lazarroz FL cultivar became a challenge and an opportunity in comparison with our previously results from CR-5272 for the following reasons. First, the alternative immersion method allowed our material to fully generate homogeneous and predictable in vitro plants after irradiation and was demonstrated to work for our Lazarroz FL cultivar and other recalcitrant rice lines CR-1821 and CR-1113. An immersion system is a little-explored tool for regeneration of gamma radiation mutated materials with great potential. The constant liquid and airflow can dilute oxidation compounds and facilitate more homogeneous exposure to nutrients. We validated the results with recalcitrant cultivars CR-1821 and CR-1113, which showed a total regeneration rate which contrasted with null regeneration in the conventional semisolid method. Plant cultivars' nonhomogeneous in vitro behavior is not fully understood, but recent discoveries provide insights into the genetic basis of sucrose metabolism and oxidation. External phytohormones seem to trigger rice sucrose metabolism required for regeneration. The system appears to rely on the expression of endogenous cytokinin, auxin, and ABA signaling genes: ORYZA SATIVA RESPONSE REGULATOR 1 (ORR1), PIN-formed 1 (PIN1), and late embryogenesis-abundant 1(LEA-1)[26]. The expression of OsSRO1c, a regulator of oxidative stress, seems to be vital for avoiding callus browning in indica cultivars [27]. Tissue culture and gamma radiation produce oxidation via reactive oxygen species (ROS), which are usually contained in chloroplasts, peroxisomes, and mitochondria. ROS include superoxide O2-, hydroxyl OH radicals and H2O2 [28]. Tissue culture and gamma rays trigger ROS, consequently damaging the DNA by oxidation of the molecule into 8-oxo-7-hydroxyguanosine (8-oxo-dG) and further transversions of C/G and T/A[29].

Second, we can develop desired mutations in a cultivar that farmers are already using, leading to faster breeding and adoption of a derived improved cultivar. Rice traits associated with specific genes are well known, which paves the way for producing novel cultivars based on mutation of desired or required conditions, such as biotic and abiotic stress tolerance [11]. We foresee developing new traits such as NaCl, herbicide, and pH tolerance based on the corresponding selection agents and not limited by recalcitrant cultivars, which used to be our bottleneck to bring innovation.

Author Response

Thank you very much for all the inputs.
We have improved the document in content and format. Your specific suggestions were incorporated as follows.

Temporary immersion system improves regeneration of in vitro irradiated recalcitrant rice embryogenic calli.

Alejandro Hernández-Soto Jason Pérez-Chavez, Rebeca Fait-Zuñiga, Randall Rojas-Vásquez, Andrés Gatica- Arias, Ana Abdelnour-Esquivel

Revisions

  1. There are multiple typographic errors. Capital letters without a razon, words without a space, etc.

Accepted. The document was reviewed by a professional service provided by AJE.

  1. The legends of the figures are very brief. They should be much more explanatory. They must be readable independently of the text.

Accepted and corrected.

  1. The discussion is very brief.

Accepted and corrected.

  1. English requires a thorough revision. There is a mixture of the tense of the verbs throughout the document.

Accepted and corrected.

  1. Electromagnetic radiation produces random, undirected mutations. It is very speculative. How will they generate the desired mutations?

Accepted and corrected.

Plant tissue culture represents an opportunity to overcome time limitations, land requirements, and selection of the desired variability that results when using gamma ray mutagenesis in seeds. Once irradiated, the seeds must be planted several times from M0=seeds prior to mutagenic treatment, M1=plants produced from material treated with the mutagen, to subsequent generations termed M2, M3, M4, to avoid chimeras and heterogenicity until exposed to stress selection conditions, as has been done in the past [6,8]. In contrast, rice tissue culture can produce a primitive cell aggregate or callus with embryogenic potential and consequently mutate and regenerate from one or a few cells with stressor selection from the beginning such as NaCl or herbicide [11,13,14].

  1. How was the dose measured? It is unclear which radiation source they used (machine) other than the radioisotope being 60 What was the geometric arrangement to ensure that all the seeds received the same dose?

Accepted and corrected. The irradiation was on calli alone.

4.4. Gamma irradiation

Embryogenic calli irradiation was achieved with a CoS 44HH-N Ob-Servo Ignis with 24 cobalt 60 sources (Institute of Isotopes Co, Ltd., Budapest, Hungary).To determine radiosensitivity and the median lethal dose (LD50), calli were irradiated at 0, 40, 60, 80 Gy. Ten repetitions, and 20 embryogenic calli per exposure were used. Survival rate was recorded after calli were transferred to regeneration medium composed by basal regeneration medium supplemented with 0.5 mg L-1 NAA + 3 mg L-1 6-BA, selected after previous experiments were analyzed. Culture conditions were the same indicated above and after 4 weeks lethal doses were calculated using probit analysis on IBM SPSS version 27 based on callus death rates.

.

  1. Sections 5 and 6 should be eliminated from the document, like Data Availability Statement & Acknowledgments. I am not sure if the authors do not really have a conflict of interest or left the journal format. The three paragraphs before the references should also be removed.

Accepted, and eliminated.

  1. In the case of supplementary material, in addition to the table, are there videos and figures? If this is not the case, it should be removed from the header. The supplementary table should be attached as a separate file and not in the body of the article.

Accepted, and eliminated.

  1. Line 15 60Co must ve 60

Accepted, and adjusted

  1. Line 59. Abbreviations must be written complete the first time.

Accepted, and adjusted

  1. Line 72. Ilalics.

Accepted, and adjusted

  1. Figure 1. It is not referenced in the text.

Accepted, and adjusted

The chloroplast maturase gene K (matK) and ribulose-1,5-bisphosphate carboxylase small/large subunit (rbcL), MZ558335 and MZ558334 sequences of Lazarroz FL showed perfect matches with the already published NCBI O. sativa indica accessions and demonstrated the identity of the non-irradiated material as expected. We detected three synonymous mutations in the rbcL sequence that are important for the characterization of the variety (Fig 1).

  1. Figure 2. Part B is out of focus, and what the authors want to see is not appreciated. It is not referenced in the text.

Accepted, and adjusted

Figure 2. Calli induction of Lazarroz FL cultivar on MS medium with 2 mg L-1 2,4-D after 30 days days of culture under darkness. (A) Embryogenic calli obtained from the scutellum of mature zygotic embryos (arrow) (B) the compact and friable calli as observed under the stereoscope.

Rice embryogenic calli obtained from induction medium supplemented with 2 mg L-1 were obtained after 30 days of culture from the scutellum of mature zygotic embryos (Fig 2A) and were composed of yellow friable aggregates (Fig 2B).

  1. Table 1. It is not referenced in the text.

Accepted, and adjusted

The embryogenic calli induction in Lazarroz FL was affected by the combination of plant growth regulators used. Thus, induction percentages ranging from 12.77 to 71.44 were obtained The higher callus induction percentage (71.44) was achieved with 2 mg L-1 2,4-D alone (Table 1). In our case, the positive induction at 2 mg L-1 of 2,4-D contrasted with higher oxidation (3.55%) and represented a challenge for the next steps of regeneration and our final irradiation goal, which also triggers oxidation (Table 1).

  1. Table 2. A comma has been used in some of the figures instead of a period. This also happens in the text.

Accepted, and adjusted

  1. Figures 3, 4, These figures are not referenced in the text.

Accepted, and adjusted

  1. There is no table 5. Table 6 is not referenced in the text.

Accepted, and adjusted

  1. Line 192. Part of the line is in a different font size.

Accepted, and adjusted

  1. Line 197. The nomenclature is wrong as the subscripts were not placed. Line 209. MS must be cited properly.

Accepted, and adjusted

  1. Line 213. 6-Benzylaminopurine (6-BAP) is an incorrect name for 6- benzyladenine (BA).

Accepted, and adjusted

  1. Line 219. The symbol for degree is missing. Must be °C. Line 223. μmol·s-1·m-2, must be μmol · s-1 m-2.

Accepted, and adjusted

Cultures were maintained at an irradiance of 72 µmol s-1m-2, a 16hrs light/8 hrs dark photoperiod was used and 26 ± 2 °C for a period of 4 weeks.

  1. References must be corrected for typographical errors, such as the one present in line 331.

Accepted, and adjusted

  1. References are not in the format requested by the journal.

Accepted, and adjusted

Reviewer 3 Report

Dear Authors,

Review of the manuscript entited „Temporary immersion system improves regeneration of in vitro irradiated recalcitrant rice embryogenic calli.” written by Alejandro Hernández-Soto et al..

The development of gamma rays mutant rice lines would be a solution for introducing variability in already farmer using varieties. In vitro gamma (60Co) mutagenesis reduces chimeras and allows a faster selection of desired traits but requires laboratory process optimization. The objective of the present work was the in vitro establishment of a recalcitrant rice embryogenic calli, the determination of its sensitivity to Co-60 gamma radiation, sequencing MATK and Rubisco genes for identification purposes, as well as generation optimization. The radiosensitivity of embryogenic calli resulted in an LD50 of 110Gy, while the 20% lethal dose was 64Gy. All sequenced genes matched perfectly with already reported MATK and Rubisco O. sativa genes with a clear SNP that  identifies the local variety related to the southeast Asia Region. Callus induction improved with an MS medium with 2mg/L 2,4D, and the regeneration was achieved with an MS medium with 3mg/L BAP and 23 0,5mg/L NAA. The optimized radiation condition was 60Gy with an 83% regeneration in a semisolid medium, allowing a balance between mutation and regeneration. An immersion system (RITA® ) of either 60 or 120 seconds in every 8 hour allowed a systematic and homogeneous total regeneration of the recalcitrant line, in contrast with the semisolid medium that resulted in positive but irregular regeneration. Other well-known recalcitrant cultivars, CR1821, CR1113 also had an improving regeneration in the immersion system, demonstrating its potential use for recalcitrant materials.

        I recommend this MS for acceptance after major revision. The list of my recommendations are presented below.

Abstract: The abstract is 263 words (according to the „Instructions for Authors for Plants” it can be max. 250 words), so it shall have to shorten.  

Additional notes:

line 18: „…sequencing MATK and Rubisco for identification purposes,”. Since here you first mention the name of MATK and Rubisco, you shall have specify their names e.g. ribulose-1,5-bisphosphate carboxylase small/large subunit (Rubisco), and the same for MATK. An other suggestion that you may prepare an Abbreviation paragraph for the several chemical namesto avoid the redundancy.

lines 20-22: „All sequenced genes matched perfectly with already reported MATK and Rubisco O. sativa genes with a clear SNP that identifies the local variety related to the southeast Asia Region.”. It is the same as for line18 in the case of SNP, and the species names shall have to write in italics allover in the text.

line 22:” Callus induction improveds  with an..” until now you used past tense, you shall have to unify the text like this.

lines 26-27: „An immersion system (RITA® ) of either 60 or 120 seconds in every 8 hour allowed a systematic and homogeneous total regeneration of the recalcitrant line.”

1.Introduction

line 39: you shall have to put the reference numbers into squared brackets in the whole text.

line 49: „Once irradiated, seeds must be planted several times M0 to M4 to avoid…” what does it mean: M0 to M4 ? describe it in detailed, please.

lines 50-51: „as done in the past [6,8].”

lines 59-61: „.Embryogenic cells result when exposed to Murashige and Skoog (MS) (1962) medium [18] supplemented with 2 to 2.5 mg/l 2,4-D  resulting in pro-embryos and somatic embryos [17–18].”

line 63: „CR-1113 and CR-5272 have predictable induction and regeneration under 2,5mg/L of 2,4D [19] .

lines 64-65: „Here we faced the challenge of identifying our material with commonly used molecular markers MATK and Rubisco,”  What are these molecular markers? please, describe MATK and Rubisco correctly in the Mat&Methods because I did not find this description there.  

2.Results

lines71-82: While you speak about the  MATK and Rubisco marker sequencing, there is not cited anywhere the Fig.1. here in the text!

line 89: „Rice tissue culture”  - here you shall have to write some senteces to introduce the rice tissue culture and in this case you can refer to the Fig.2., which is otherwise missing ! Or alternatively, you can put the Fig.2. into the Supplementary material.

line 95: in generally, all the Tables 1.-6. titles  you shall have to put under the Tables. Especially, in the case of Table 1., the wholeTable 1. shall have to put under the line 102.

Additional note: how did you measure the oxidation rate, for this I did not find any description in the Math&Methods, this shall have to replace.

line 107: the Table 2. title you shall have to put under the Table 2., into line 109.

What does it mean the ANA in the Table 2., I did not find its description in the Math&Methods.  

line 115: Try to introduce the gamma rays irradiation by some sentences and after that represent the results of this method by Table 3.

line 117: the Table 3. title you shall have to put under the Table 3., into line 118.

line 127: Try to introduce the Figure 3. by some sentences because this missing very much, the text is very confused like the present form or put the lines 131-132 over the Fig.3 and here you can cite the Fig.3.

lines133-134: Table 6. is only Table 5., if you calculate well. The Table 5. title you shall have to put under the Table 5., into line 134. An other note that I did not find the description of the temporal immersion system anywhere, which is also called in this manuscrip as RITA, and this is not mentioned in the Math&Methods. This should be important to describe there. My other problem is that in the Table 5. (previously Table 6.) the immersion time per 8hrs, while in the Table 6. (previously Table 7.) the immersion time per 12 hrs is written. Sincet here is no description for immersion method,  such I do not understand why do you use two different immersion methods?

line 136: again, here you should introduce the Fig.4. by some sentences, otherwise there is not exist citation for the Figure 4.in the text (at this moment you do not cite this Fig.4.at all).

Line 144: the Table 6. (previously Table 7.) title you shall have to put under the Table 6., into line 145.

Discussion:

line 166: here you mention that the „Regeneration was improved using a temporal immersion system (RITA® ) with either 60 and 120 seconds of immersion every 8hours achieving a predictable and more homogeneous regeneration response”. This temporal immersion system (RITA®) is what you shall have to describe in the MAth&Methods carefully.

lines 191-192:” The system appears to rely on the expression of endogenous cytokinin, auxin, and abscissic acid (ABA ), signaling genes: ORR1, PIN-formed 1 (PIN1), and late embryogenesis-abundant 1(LEA-1).” You shall have to specify the ORR1 also.

line 196:The tissue culture and gamma radiation produce oxidation via reactive oxygen spe- 195 cies (ROS) usually contained in the chloroplast, peroxisomes, and mitochondria.” Here is a reference is missing…

line 197: „ROS includes superoxide (O2-), hydroxyl radical (OH.) radical, hydroxil peroxide (H2O2) and HO2 (what is this? such ROS does not exists!).

line 199: the numbering of the References not consequent because here you refer to the Ref. 25 and this is the last reference in the text and there is 27 references. Correct this, please. Additonal note that what the SIT means here? You shall have to specify this also.

Mat&Methods:

line 223: „…, photoperiod 16h and temperature 26 ± 2 °C.” Here the photoperiod means that there is 16hrs light and 8 hrs dark for callus cultures? In this case you may write that 16hrs light/8 hrs dark photoperiod was used.

lines 229-230: „Kit NucleoSpinTM Tissue (Macherey-Nagel) used for DNA extraction and Thermo Fisher K1071 for subsequent PCR following the recommendations of the provider.” Here you shall have to specify the producer correctly for DNA isolation and PCR kits like Kit NucleoSpinTM Tissue (Macherey-Nagel, firm name, city name, state name where from it was purchesed) and the same for the Thermo Fisher K1071 PCR kit.

Other deficiencies from the Math&Methods: I miss the description how the temporal immersion tests were carried out and how the oxidative damage was measured after cobald irradiation?

I also did not find any citations for the Supplemental materials, this is completely missing from the text.

References:

General note for Refrences: you shall have to shorten the cited journal names.

line 324: in the Ref.15. the journal name, volume and pages are missing.

Sincerely yours,

Reviewer 1

Author Response

Thank you very much for all the inputs.
We have improved the document in content and format. Your specific suggestions were incorporated as follows.

Review of the manuscript entited „Temporary immersion system improves regeneration of in vitro irradiated recalcitrant rice embryogenic calli.” written by Alejandro Hernández-Soto et al..

The development of gamma rays mutant rice lines would be a solution for introducing variability in already farmer using varieties. In vitro gamma (60Co) mutagenesis reduces chimeras and allows a faster selection of desired traits but requires laboratory process optimization. The objective of the present work was the in vitro establishment of a recalcitrant rice embryogenic calli, the determination of its sensitivity to Co-60 gamma radiation, sequencing MATK and Rubisco genes for identification purposes, as well as generation optimization. The radiosensitivity of embryogenic calli resulted in an LD50 of 110Gy, while the 20% lethal dose was 64Gy. All sequenced genes matched perfectly with already reported MATK and Rubisco O. sativa genes with a clear SNP that  identifies the local variety related to the southeast Asia Region. Callus induction improved with an MS medium with 2mg/L 2,4D, and the regeneration was achieved with an MS medium with 3mg/L BAP and 23 0,5mg/L NAA. The optimized radiation condition was 60Gy with an 83% regeneration in a semisolid medium, allowing a balance between mutation and regeneration. An immersion system (RITA® ) of either 60 or 120 seconds in every 8 hour allowed a systematic and homogeneous total regeneration of the recalcitrant line, in contrast with the semisolid medium that resulted in positive but irregular regeneration. Other well-known recalcitrant cultivars, CR1821, CR1113 also had an improving regeneration in the immersion system, demonstrating its potential use for recalcitrant materials.

        I recommend this MS for acceptance after major revision. The list of my recommendations are presented below.

Abstract: The abstract is 263 words (according to the „Instructions for Authors for Plants” it can be max. 250 words), so it shall have to shorten.  

R/. Accepted. The abstract is 250 words.

Abstract: The development of gamma ray-mutated rice lines is a solution for introducing genetic variability in indica rice varieties already being used by farmers. In vitro gamma ray (60Co) mutagenesis reduces chimeras and allows a faster selection of desired traits but requires optimization of the laboratory procedure. The objectives of the present work were sequencing of matK and rbcL genes for identification purposes, the in vitro establishment of recalcitrant rice embryogenic calli, the determination of their sensitivity to gamma radiation, and optimization of the generation procedure. All sequenced genes matched perfectly with previously reported matK and rbcL O. sativa genes, with a clear SNP that identified the local variety Lazarroz FL as being related to one in the Southeast Asia region. Embryogenic callus induction improved in MS medium containing 2 mg L-1 2,4-D, and regeneration was achieved with MS medium with 3 mg L-1 BA and 0.5 mg L-1 NAA. The optimized radiation condition was 60 Gy, (LD20=64 Gy) because it maintains an 83% regeneration in a semisolid medium, allowing a balance between mutation and regeneration. An immersion system (RITA®) of either 60 or 120 seconds every 8 hours allowed systematic and homogeneous total regeneration of the recalcitrant line, in contrast with the semisolid medium that resulted in positive but irregular regeneration. Other well-known recalcitrant cultivars, CR1821 and CR1113, also had improved regeneration in the immersion system. To our knowledge, this is the first report on using an immersion system to allow regeneration of gamma-ray mutants from recalcitrant indica rice materials.

Additional notes:

line 18: „…sequencing MATK and Rubisco for identification purposes,”. Since here you first mention the name of MATK and Rubisco, you shall have specify their names e.g. ribulose-1,5-bisphosphate carboxylase small/large subunit (Rubisco), and the same for MATK. An other suggestion that you may prepare an Abbreviation paragraph for the several chemical namesto avoid the redundancy.

  1. Accepted. Adjusted in introduction as follows.

Here, we faced the challenge of identifying a local rice line Lazarroz FL currently used by farmers with the commonly used molecular markers, the chloroplast maturase gene K (matK) and ribulose-1,5-bisphosphate carboxylase small/large subunit (rbcL)

lines 20-22: „All sequenced genes matched perfectly with already reported MATK and Rubisco O. sativa genes with a clear SNP that identifies the local variety related to the southeast Asia Region.”. It is the same as for line18 in the case of SNP, and the species names shall have to write in italics allover in the text.

  1. Accepted and corrected during the text.

All sequenced genes matched perfectly with previously reported matK and rbcL O. sativa genes, with a clear SNP that identified the local variety Lazarroz FL as being related to one in the Southeast Asia region

line 22:” Callus induction improved with an..” until now you used past tense, you shall have to unify the text like this.

  1. Accepted and corrected.

Embryogenic callus induction improved in MS medium containing 2 mg L-1 2,4-D, and regeneration was achieved with MS medium with 3 mg L-1 BA and 0.5 mg L-1 NAA. The optimized radiation condition was 60 Gy, (LD20=64 Gy) because it maintains an 83% regeneration in a semisolid medium, allowing a balance between mutation and regeneration

lines 26-27: „An immersion system (RITA® ) of either 60 or 120 seconds in every 8 hour allowed a systematic and homogeneous total regeneration of the recalcitrant line.”

  1. Accepted and corrected.

An immersion system (RITA®) of either 60 or 120 seconds in every 8 hour allowed systematic and homogeneous total regeneration of the recalcitrant line, in contrast with the semisolid medium that resulted in positive but irregular regeneration.

1.Introduction

line 39: you shall have to put the reference numbers into squared brackets in the whole text.

  1. Accepted and corrected.

line 49: „Once irradiated, seeds must be planted several times M0 to M4 to avoid…” what does it mean: M0 to M4 ? describe it in detailed, please.

  1. Accepted and corrected.

Once irradiated, the seeds must be planted several times from M0=seeds prior to mutagenic treatment, M1=plants produced from material treated with the mutagen, to subsequent generations termed M2,M3,M4, to avoid chimeras and heterogenicity until exposed to stress selection conditions, as has been done in the past

lines 50-51: „as done in the past [6,8].”

  1. Accepted and corrected.

Once irradiated, the seeds must be planted several times from M0=seeds prior to mutagenic treatment, M1=plants produced from material treated with the mutagen, to subsequent generations termed M2,M3,M4, to avoid chimeras and heterogenicity until exposed to stress selection conditions, as has been done in the past [6,8].

lines 59-61: „.Embryogenic cells result when exposed to Murashige and Skoog (MS) (1962) medium [18] supplemented with 2 to 2.5 mg/l 2,4-D  resulting in pro-embryos and somatic embryos [17–18].”

  1. Accepted and corrected.

Embryogenic cells result when exposed to Murashige and Skoog (MS) medium supplemented with 2 to 2.5 mg L-1 2,4-D, resulting in pro-embryos and somatic embryos [16–18].

line 63: „CR-1113 and CR-5272 have predictable induction and regeneration under 2,5mg/L of 2,4D [19] .

  1. Accepted and corrected.

Costa Rican cultivars such as CR-201, CR-1707, CR-1821, CR-8334, and CR-8341 have unpredictable and variable behavior, while CR-1113 and CR-5272 have predictable induction and regeneration in 2.5 mg L-1 2,4-D [19].

lines 64-65: „Here we faced the challenge of identifying our material with commonly used molecular markers MATK and Rubisco,”  What are these molecular markers? please, describe MATK and Rubisco correctly in the Mat&Methods because I did not find this description there.  

  1. Accepted and corrected.

Here, we faced the challenge of identifying a local rice line Lazarroz FL currently used by farmers with the commonly used molecular markers, the chloroplast maturase gene K (matK) and ribulose-1,5-bisphosphate carboxylase small/large subunit (rbcL), improving our plant tissue culture methods, and determining the radiosensitivity of embryogenic calli and further plant regeneration.

2.Results

lines71-82: While you speak about the  MATK and Rubisco marker sequencing, there is not cited anywhere the Fig.1. here in the text!

  1. Accepted and corrected.

The chloroplast maturase gene K (matK) and ribulose-1,5-bisphosphate carboxylase small/large subunit (rbcL), MZ558335 and MZ558334 sequences of Lazarroz FL showed perfect matches with the already published NCBI O. sativa indica accessions and demonstrated the identity of the non-irradiated material as expected. We detected three synonymous mutations in the rbcL sequence that are important for the characterization of the variety (Fig 1).

line 89: „Rice tissue culture”  - here you shall have to write some senteces to introduce the rice tissue culture and in this case you can refer to the Fig.2., which is otherwise missing ! Or alternatively, you can put the Fig.2. into the Supplementary material.

  1. Accepted and corrected.

2.2 Embryogenic calli induction

[…..]

Rice embryogenic calluses obtained from induction medium supplemented with 2 mg L-1 were obtained after 30 days of culture from the scutellum of mature zygotic embryos (Fig 2A) and were composed of yellow friable aggregates (Fig 2B).

line 95: in generally, all the Tables 1.-6. titles  you shall have to put under the Tables. Especially, in the case of Table 1., the wholeTable 1. shall have to put under the line 102.

  1. Accepted and corrected.

2.2 Embryogenic calli induction

Table 1. Rice callus induction and oxidation rate from different induction treatments

Treatment

id

Embryogenic calli (%)

Oxidation rate (%)

2,5 mg 2,4-D

i

21.44 b

2.00 b

2 mg 2,4-D

ii

71.44 a

3.55 a

1 mg BA + 1 mg de 2,4-D

iii

12.77 d

0.66 c

1 mg BA + 2 mg de 2,4-D

iv

16.77 c

0.21 c

2 mg 2,4-D + 0,25 mg de TDZ

v

23.00 b

0.66 c

1 All treatments had 30 replicates of 30 seeds per replicate (n=900). Letters represent a significant difference (p ≤ 0.05).

The embryogenic calli induction in Lazarroz FL was affected by the combination of plant growth regulators used. Thus, induction percentages ranging from 12.77 to 71.44 were obtained The higher callus induction percentage (71.44) was achieved with 2 mg L-1 2,4-D alone (Table 1). In our case, the positive induction at 2 mg L-1 of 2,4-D contrasted with higher oxidation (3.55%) and represented a challenge for the next steps of regeneration and our final irradiation goal, which also triggers oxidation (Table 1).

Additional note: how did you measure the oxidation rate, for this I did not find any description in the Math&Methods, this shall have to replace.

  1. Accepted and corrected.

4.2. Embryogenic callus induction

Callus induction, oxidation rates (brown or necrotic/total calli)) were recorded as response variables and analyzed in a completely randomized design with a generalized linear model with a Poisson distribution and logit link function. Post hoc analysis consisted of an Honest Significant Difference test to establish differences between the means.

4.3 Regeneration on semisolid medium

Calli showing mature coleoptilar germinated embryos were determined as regenerated, calli with completely differentiated plantlets greater than 1 cm were categorized as sprouted, and necrotic calli with dark coloration were identified as oxidated calli.

line 107: the Table 2. title you shall have to put under the Table 2., into line 109.

  1. Table corrected and put under subtitle 2.3

2.3 Regeneration on semisolid medium

The best regeneration rate of approximately 70% resulted from 0.5 mg of NAA+ 3 mg of 6-BA, with sprouting of 7.14% and oxidation of only 9.52% (Table 2). Other regeneration medium recipes also resulted in regeneration but with higher oxidation and were consequently useless for our next step, the gamma radiation mutagenesis.

Table 2. Rice calli regeneration, sprouting and oxidation rate of different treatments

Induction

Treatment

Regeneration Treatment

Regeneration (%)

Sprouting

 (%)

Oxidation

 (%)

2 mg 2,4-D

0.5 mg NAA + 3 mg BA

69.04 a

7.14 ab

9.52 d

0.5 mg NAA + 0.5 mg TDZ

38.09 c

2.38 b

61.90 a

0.5 mg NAA + 0.5 mg Kinetin

47.61 b

9.52 a

23.80 c

0.5 mg NAA + 0.5 mg BA

28.57 d

2.38 b

54.76 b

1 mg BA + 2 mg 2,4-D

0.5 mg NAA + 3 mg BA

28.29 b

0 a

58.43 b

0.5 mg NAA + 0,25 mg TDZ

58.82 a

0 a

100 a

0.5 mg NAA + 0.5 mg Kinetin

9.22 c

0 a

56.81 b

0.5 mg NAA + 0.5 mg BA

12.82 c

0 a

44.26 c

1 mg BA + 1 mg 2,4-D

0.5 mg de NAA + 3 mg BA

58.45 ab

16.38 a

18. 69 bc

0.5 mg de NAA + 0.5 mg TDZ

61.75 a

10.71 ab

27.93 a

0.5 mg de NAA + 0.5 mg Kinetin

56.31 ab

15.92 a

20.01 ab

0.5 mg de NAA + 0.5 mg BA

49.88 b

4.16 b

11.66 c

0.5 mg de NAA + 3 mg BA

58.45 ab

16.38 a

18. 69 bc

2,5 mg 2,4-D

0.5 mg NAA + 3 mg BA

34.64 b

0 b

9.20 c

0.5 mg NAA + 0.5 mg TDZ

51.41 a

9.61 a

22.96 a

0.5 mg NAA + 0.5 mg Kinetin

43.62 ab

0 b

15.73 b

0.5 mg NAA + 0.5 mg BA

18.00 c

0 b

15.19 b

2 mg 2,4-D + 0,25 mg TDZ

0.5 mg NAA + 3 mg BA

77.27 a

2.27 a

96.59 a

0.5 mg NAA + 0.5 mg TDZ

50.25 b

0 a

86.36 b

0.5 mg NAA + 0.5 mg Kinetin

73.86 a

3.40 a

82.95 b

0.5 mg NAA + 0.5 mg BA

44.29 b

0 a

72.81 c

1 All treatments had 6 replicates of 7 calli each replicate (n=42). Letters represent significant differences (p ≤ 0.05).

What does it mean the ANA in the Table 2., I did not find its description in the Math&Methods.  

  1. Accepted and corrected. It corresponds to NAA instead.MM adjusted.

4.3 Regeneration on semisolid medium

Embryogenic calli obtained from each induction medium were transferred to different regeneration treatments. Basal regeneration medium was similar to that described by Sudhakar et al. (1998)28, and was constituted by MS mineral salts and vitamins, 20 g.L-1 sucrose and 0.3 g.L-1 hydrolyzed casein. Treatments consisted on supplementing the basal medium with variations on growth regulators: control was supplemented with 0.5 mg L-1 α-Naphthaleneacetic Acid (NAA) + 3 mg L-1 6-BA, second treatment was supplemented with 0.5 mg L-1 NAA + 0.5mg.L-1 6-BA, third treatment was supplemented with 0.5 mg L-1 NAA + 1.5 mg L-1 Kinetin (KIN) and a fourth treatment consisted on supplementing basal medium with 0.5 mg L-1 NAA + 0.5 mg L-1 TDZ. After adding growth regulators, pH was adjusted to 5.8 with 1 N NaOH or 1 N HCl and 5.4 g L-1 Gelzan ® were added as gelling agent. All previously mentioned reagents were supplied by Phytotechnology Laboratories. After a properly dissolving of gelling agent, 60 mL of media were dispensed on 475 mL polypropylene WNA Deli Containers, and afterwards, media were sterilized at 1.2 ATM.cm-2 and 121 C for 30 min. For each treatment, 6 replicates of 7 calli were cultured on a factorial design. Factor 1 consisted of induction media and factor 2 was regeneration media. Cultures were maintained at an irradiance of 72 µmol s-1m-2, a photoperiod of 16 h and 26 ± 2 C for a period of 4 weeks. Calli showing mature coleoptilar germinated embryos were determined as regenerated, calli with completely differentiated plantlets greater than 1 cm were categorized as sprouted, and necrotic calli with dark coloration were identified as oxidated calli. These response variables were analyzed with a generalized linear model with a Poisson distribution and logit link function. Post hoc analysis consisted of an Honest Significant Difference test and differences between each combination of factors were recognized.

line 115: Try to introduce the gamma rays irradiation by some sentences and after that represent the results of this method by Table 3.

line 117: the Table 3. title you shall have to put under the Table 3., into line 118.

  1. Accepted and corrected.

2.4 Gamma radiation mutagenesis

The effect of cobalt-60 (60Co) gamma radiation on embryogenic indica rice callus was evaluated. The lethal dose (LD50) of the embryogenic calli was 110 Gy, while the 20% lethal dose was 64 Gy, resulting in a 0 to 120 gray gradient exposure, with 200 calli per exposure (Table 3).

Table 3. Lethal effect of gamma radiation on the embryogenic rice calli1

Lethal gamma rays

dose model

Dose (Gy)

Lower limit (Gy)

Upper limit (Gy)

LD10

41,145

34,552

46,708

LD20

64,799

60,083

69,20

LD25

73.785

69,388

78,139

LD30

81,855

77,507

86,403

LD40

96,429

91,649

101,85

LD50

110,050

104,435

116,720

1 All treatments had 10 replicates with 20 calli each (n=200), p ≤ 0.05. Data were compiled at 30 days post-radiation.

line 127: Try to introduce the Figure 3. by some sentences because this missing very much, the text is very confused like the present form or put the lines 131-132 over the Fig.3 and here you can cite the Fig.3.

  1. Accepted and corrected.

Gamma-irradiated calli regenerated plants after 45 days in regeneration medium (Fig 3A) and fully in vitro plants (Fig 3B) at 60 days post-irradiation at 60 Gy.

Figure 3. Regeneration of 60 Gy irradiated calli Lazarroz FL cultivar after (A) 45 days and (B) 60 days.

lines133-134: Table 6. is only Table 5., if you calculate well. The Table 5. title you shall have to put under the Table 5., into line 134. An other note that I did not find the description of the temporal immersion system anywhere, which is also called in this manuscrip as RITA, and this is not mentioned in the Math&Methods. This should be important to describe there. My other problem is that in the Table 5. (previously Table 6.) the immersion time per 8hrs, while in the Table 6. (previously Table 7.) the immersion time per 12 hrs is written. Sincet here is no description for immersion method,  such I do not understand why do you use two different immersion methods?

  1. Accepted and corrected.

The potential of a temporary immersion system for regenerating recalcitrant materials (CR-5272, CR-1821, CR-1113 and Lazarroz FL) was evaluated. Our results showed no significant differences between the immersion time (60 vs 120 s) on the regeneration capacity, sprouting and oxidation (Table 5). Nevertheless, a higher regeneration rate was obtained using the RITA® with 60 s (100%) compared to regeneration on the best semi-solid medium (70 %). Similarly, sprouting rate was higher using the RITA® system independently of the immersion time compared to that obtained using the selected semi-solid medium (Table 2 and 5).

Although, the immersion time (60 vs 120 s) did not significantly affect the oxidation rate, it was higher using an immersion of 120 s (97.56%) compared to 60 s (60.00 %) and semi-solid medium (9.52%) (Table 2 and 5).

Table 5. Immersion regeneration, sprouting and oxidation rates1

Immersion time

Regeneration

Sprouting

Oxidation

60 seconds

100.00 a

25.00 a

60.00 a

120 seconds

97.56 a

31.71 a

97.56 a

1 All treatments had 4 replicates of 10 calli per replicate (n=40), p ≤ 0.05. Data were compiled at 15 days post-radiation.

The potential of a temporary immersion system for regenerating recalcitrant materials was also validated for the well-known recalcitrant cultivars CR-1821 and CR-1113 (Fig. 5).

Figure 5. Regeneration of cultivars CR-5272 and recalcitrant CR-1821 and CR-1113 in MS medium with 0.5 mg L-1 of NAA + 3 mg L-1 of BA from calli induced with 2 mg L-1 of 2,4-D. (A) CR-5272 regeneration in RITA with immersion for 30 seconds or 60 seconds, and the semisolid medium control. (B) CR-1821 regeneration in RITA with immersion for 30 seconds or 60 seconds and the semisolid medium control. (C) CR-1113 regeneration in RITA with immersion for 30 seconds or 60 seconds and the semisolid medium control. Note that all RITA treatments contain calli with green areas that regenerate into plants. The absence of regeneration of the recalcitrant cultivars CR-1821 and CR-1113 was observed in the semisolid media.

line 136: again, here you should introduce the Fig.4. by some sentences, otherwise there is not exist citation for the Figure 4.in the text (at this moment you do not cite this Fig.4.at all).

Line 144: the Table 6. (previously Table 7.) title you shall have to put under the Table 6., into line 145.

  1. Accepted and corrected.

Discussion:

line 166: here you mention that the „Regeneration was improved using a temporal immersion system (RITA® ) with either 60 and 120 seconds of immersion every 8hours achieving a predictable and more homogeneous regeneration response”. This temporal immersion system (RITA®) is what you shall have to describe in the MAth&Methods carefully.

  1. Accepted and corrected.

4.5. Regeneration in Recipient for Automated Temporary Immersion (RITA®)

The RITA® temporary immersion system regeneration of the embryogenic calli consisted of 200 mL regeneration media previously described in 4.3, with 23 four-week-old calli per unit, different rice cultivars (Lazarroz FL, CR-5272, CR-1821, and CR-1113). Light and temperature culture conditions remained unchanged, with 60 or 120 seconds immersion as treatments, every eight hours. After four weeks of culture, the variables evaluated were regeneration, sprouting, and oxidation, green area calculations with ImageJ version 1.52p.

lines 191-192:” The system appears to rely on the expression of endogenous cytokinin, auxin, and abscissic acid (ABA ), signaling genes: ORR1, PIN-formed 1 (PIN1), and late embryogenesis-abundant 1(LEA-1).” You shall have to specify the ORR1 also.

  1. Accepted and corrected.

The system appears to rely on the expression of endogenous cytokinin, auxin, and ABA signaling genes: ORYZA SATIVA RESPONSE REGULATOR 1 (ORR1), PIN-formed 1 (PIN1), and late embryogenesis-abundant 1(LEA-1)[25]. The expression of OsSRO1c, a regulator of oxidative stress, seems to be vital for avoiding callus browning in indica cultivars [26].

line 196: „The tissue culture and gamma radiation produce oxidation via reactive oxygen spe- 195 cies (ROS) usually contained in the chloroplast, peroxisomes, and mitochondria.” Here is a reference is missing…

line 197: „ROS includes superoxide (O2-), hydroxyl radical (OH.) radicalhydroxil peroxide (H2O2) and HO2 (what is this? such ROS does not exists!).

  1. Accepted and corrected.

Tissue culture and gamma radiation produce oxidation via reactive oxygen species (ROS), which are usually contained in chloroplasts, peroxisomes, and mitochondria. ROS include superoxide O2-, hydroxyl OH radicals and H2O2 [27].

  1. Debnath S.; Chandel R.K.; Devi K.; Khan Z. Mechanism and Molecular Response of Induced Genotoxicity and Oxidative Stress in Plants. In: Khan Z., Ansari M.Y.K., Shahwar D. (eds) Induced Genotoxicity and Oxidative Stress in Plants 2021. Springer, Singapore. https://doi.org/10.1007/978-981-16-2074-4_8

line 199: the numbering of the References not consequent because here you refer to the Ref. 25 and this is the last reference in the text and there is 27 references. Correct this, please.

  1. Accepted and corrected. Last reference updated in MM [29]

4.3 Regeneration on semisolid medium

Embryogenic calli obtained from each induction medium were transferred to different regeneration treatments. Basal regeneration medium was similar to that described by Sudhakar et al. [29]

 Additonal note that what the SIT means here? You shall have to specify this also.

  1. Accepted and corrected.
  2. Conclusions

Rice tissue culture is a tool for conventional and modern breeding but is limited to the genotype response, particularly during the regeneration of recalcitrant varieties. Our results of using an immersion system help overcome such difficulties and allow the induction of gamma-ray mutants. Temporary immersion system seems to help overcome oxidation while allowing the tissue to recover and consequently becoming a more efficient method. We foresee that having access to such methods could diminish the time to trigger innovation and focus on selecting mutants with desired traits in commercially used varieties.

Mat&Methods:

line 223: „…, photoperiod 16h and temperature 26 ± 2 °C.” Here the photoperiod means that there is 16hrs light and 8 hrs dark for callus cultures? In this case you may write that 16hrs light/8 hrs dark photoperiod was used.

  1. Accepted and corrected.

Cultures were maintained at an irradiance of 72 µmol s-1m-2, a 16hrs light/8 hrs dark photoperiod was used and 26 ± 2 C for a period of 4 weeks.

lines 229-230: „Kit NucleoSpinTM Tissue (Macherey-Nagel) used for DNA extraction and Thermo Fisher K1071 for subsequent PCR following the recommendations of the provider.” Here you shall have to specify the producer correctly for DNA isolation and PCR kits like Kit NucleoSpinTM Tissue (Macherey-Nagel, firm name, city name, state name where from it was purchesed) and the same for the Thermo Fisher K1071 PCR kit.

  1. Accepted and corrected.

4.1. Molecular markers

A NucleoSpinTM Tissue Kit (Macherey-Nagel, Germany) was used for DNA extraction from 1 mg of on non-irradiated lyophilized leaf tissue of Lazarroz FL. Thermo Fisher K1071 was used for the subsequent PCR following the recommendations of the manufacturer.

Other deficiencies from the Math&Methods: I miss the description how the temporal immersion tests were carried out and how the oxidative damage was measured after cobald irradiation?

  1. Accepted and corrected.

4.2. Embryogenic callus induction

Callus induction, oxidation rates (brown or necrotic/total calli)) were recorded as response variables and analyzed in a completely randomized design with a generalized linear model with a Poisson distribution and logit link function. Post hoc analysis consisted of an Honest Significant Difference test to establish differences between the means.

4.3 Regeneration on semisolid medium

Calli showing mature coleoptilar germinated embryos were determined as regenerated, calli with completely differentiated plantlets greater than 1 cm were categorized as sprouted, and necrotic calli with dark coloration were identified as oxidated calli.

I also did not find any citations for the Supplemental materials, this is completely missing from the text.

  1. Accepted and corrected. Supplemental materials were deleted.

References:

General note for Refrences: you shall have to shorten the cited journal names.

line 324: in the Ref.15. the journal name, volume and pages are missing.

  1. Accepted and corrected. Reference changed.

 Binte Mostafiz, S.; Wagiran, A. Efficient Callus Induction and Regeneration in Selected Indica Rice. Agronomy 20188, 77. https://doi.org/10.3390/agronomy8050077

Sincerely yours,

Reviewer 1

Round 2

Reviewer 2 Report

The new version of the manuscript is fine for publication.

Author Response

Thanks for reviewing the document. We adjusted the English language and style as requested.

Reviewer 3 Report

Dear Authors,

Second review of the manuscript entited „Temporary immersion system improves regeneration of in vitro irradiated recalcitrant rice embryogenic calli.” written by Alejandro Hernández-Soto et al..

The manuscript have improved very well, so I can recommend it for acception after correcting really some mistakes. The list of the required corrections are below:

Abstract:

I am really sorry that I also had a mistake during reviewing of your manuscript, especially concerning the Abstract. The Abstract according to the „Instructions for Authors for Plants” can be only max. 200 words, not 250 words, therefore your Abstract shall have to still shorten to 200 words. I apologize for this mistake from you all.  Please, try to correct this again.

Discussion:

line 977: „ROS include superoxide (O2 -) , hydroxyl radical (OH.) and hydrogen peroxide (H2O2) [28].

Mat&Methods:

paragraph 4.1. : for the Macherey-Nagel and Thermo Fisher K1071 PCR kit the city name and the state name where from they were purchesed is still missing

lines 1248-1249: „. Post hoc analysis consisted of an Honest Significant Difference test to establish differences between the means [here is a references or website is necessary to cite for this] .

line 1457: „All previously mentioned reagents were supplied by Phytotechnology Laboratories ( )”.  The city name and the state name where from they were purchesed is missing.

line 1480: „using probit analysis on IBM SPSS version 27 (here is a website citation for this is missing) based on callus death rates.

line 1488: „sprouting, and oxidation, green area calculations with ImageJ version 1.52p (here is a website citation for ImageJ version 1.52p is missing).

References:

General note for Refrences: you still shall have to shorten the cited journal names like:

Ref. 12. Oladosu, Y.; Rafii, M. Y.; Abdullah, N.; Hussin, G.; Ramli, A.; Rahim, H. A.; Miah, G.; Usman, M. Principle and Application of Plant 1560 Mutagenesis in Crop Improvement: A Review. Biotech. Biotechnol. Equipment 2016, 30 (1), 1–16.

or

Ref.18: Radziah C.M.Z., C.; Naji Alhasnawi, A.; A. Kadhimi, A.; Isahak, A.; Mohamad, A.; Farshad Ashraf, M.; Doni, F.; Mohtar Wan Yusoff, W. Development of a Technique for Callus Induction and Plant Regeneration in Oryza sativa L. Var. MRQ74 and MR269. Advance J. Food Sci. Technol. 2017, 13 (3), 128–137.

Sincerely yours,

Reviewer 1

Author Response

Abstract:

I am really sorry that I also had a mistake during reviewing of your manuscript, especially concerning the Abstract. The Abstract according to the „Instructions for Authors for Plants” can be only max. 200 words, not 250 words, therefore your Abstract shall have to still shorten to 200 words. I apologize for this mistake from you all.  Please, try to correct this again.

Accepted and adjusted (198 words).

The development of gamma ray-mutated rice lines is a solution for introducing genetic variability in indica rice varieties already being used by farmers. In vitro gamma ray (60Co) mutagenesis reduces chimeras and allows a faster selection of desired traits but requires optimization of the laboratory procedure. The objectives of the present work were sequencing of matK and rbcL, the in vitro establishment of recalcitrant rice embryogenic calli, the determination of their sensitivity to gamma radiation, and optimization of the generation procedure. All sequenced genes matched perfectly with previously reported matK and rbcL O. sativa genes. Embryogenic calli induction improved in MS medium containing 2 mg L-1 2,4-D, and regeneration was achieved with MS medium with 3 mg L-1 BA and 0.5 mg L-1 NAA. The optimized radiation condition was 60 Gy, (LD20=64 Gy) with 83% regeneration. An immersion system (RITA®) of either 60 or 120 seconds in every 8-hour allowed systematic and homogeneous total regeneration of the recalcitrant line. Other well-known recalcitrant cultivars, CR1821 and CR1113, also had improved regeneration in the immersion system. To our knowledge, this is the first report on using an immersion system to allow regeneration of gamma-ray mutants from recalcitrant indica rice materials.

Discussion:

line 977: „ROS include superoxide (O-) , hydroxyl radical (OH.) and hydrogen peroxide (H2O2) [28].

 Accepted and adjusted,

Tissue culture and gamma radiation produce oxidation via reactive oxygen species (ROS), usually contained in chloroplasts, peroxisomes, and mitochondria. ROS include superoxide (O2-), hydroxyl (OH-) radicals, and hydrogen peroxide (H2O2) [28].

Mat&Methods:

paragraph 4.1. : for the Macherey-Nagel and Thermo Fisher K1071 PCR kit the city name and the state name where from they were purchesed is still missing

 Accepted and adjusted.

A NucleoSpinTM Tissue Kit Macherey-Nagel (Düren, Germany) was used for DNA extraction from 1 mg of on non-irradiated lyophilized leaf tissue of Lazarroz FL. Thermo Fisher K1071 (Vilnius, Lithuania) was used for the subsequent PCR following the recommendations of the manufacturer

lines 1248-1249: „. Post hoc analysis consisted of an Honest Significant Difference test to establish differences between the means [here is a references or website is necessary to cite for this] .

 Accepted and adjusted.

Post hoc analysis consisted of an Honest Significant Difference test on IBM SPSS version 27 [30] and differences between each combination of factors were recognized.

  1. IBM Corp. Released 2020. IBM SPSS Statistics for Windows, Version 27.0. Armonk, NY: IBM Corp.

line 1457: „All previously mentioned reagents were supplied by Phytotechnology Laboratories ( )”.  The city name and the state name where from they were purchesed is missing.

Accepted and adjusted.

Phytotechnology Laboratories® (Shawnee Mission, Kansas, USA)

line 1480: „using probit analysis on IBM SPSS version 27 (here is a website citation for this is missing) based on callus death rates.

 Accepted and adjusted. Ref [30]

  1. IBM Corp. Released 2020. IBM SPSS Statistics for Windows, Version 27.0. Armonk, NY: IBM Corp.

line 1488: „sprouting, and oxidation, green area calculations with ImageJ version 1.52p (here is a website citation for ImageJ version 1.52p is missing).

Accepted and adjusted

After four weeks of culture, the variables evaluated were regeneration, sprouting, and browning, green area calculations with ImageJ version 1.52p [32].

  1. Schneider, C., Rasband, W. & Eliceiri, K. NIH Image to ImageJ: 25 years of image analysis. Nat. Methods 2012, 9, 671–675. https://doi.org/10.1038/nmeth.2089

References:

General note for Refrences: you still shall have to shorten the cited journal names like:

Ref. 12. Oladosu, Y.; Rafii, M. Y.; Abdullah, N.; Hussin, G.; Ramli, A.; Rahim, H. A.; Miah, G.; Usman, M. Principle and Application of Plant 1560 Mutagenesis in Crop Improvement: A Review. Biotech. Biotechnol. Equipment 2016, 30 (1), 1–16.

or

Ref.18: Radziah C.M.Z., C.; Naji Alhasnawi, A.; A. Kadhimi, A.; Isahak, A.; Mohamad, A.; Farshad Ashraf, M.; Doni, F.; Mohtar Wan Yusoff, W. Development of a Technique for Callus Induction and Plant Regeneration in Oryza sativa L. Var. MRQ74 and MR269. Advance J. Food Sci. Technol. 201713 (3), 128–137.

Accepted and adjusted

  1. Ali, J.; Nicolas, K.L.C.; Akther, S.; Torabi, A.; Ebadi, A.A.; Marfori-Nazarea, C.M.; Mahender, A. Improved Anther Culture Media for Enhanced Callus Formation and Plant Regeneration in Rice (Oryza Sativa L.). Plants 2021, 10, 839, doi:3390/plants10050839.
  2. Wing, R. A.; Purugganan, M. D.; Zhang, Q. The Rice Genome Revolution: From an Ancient Grain to Green Super Rice. Rev. Genet. 2018, 19 (8), 505–517. https://doi.org/10.1038/s41576-018-0024-z.
  3. Chen, E.; Huang, X.; Tian, Z.; Wing, R. A.; Han, B. The Genomics of Oryza Species Provides Insights into Rice Domestication and Heterosis. Rev. Plant Biol. 2019, 70, 639–665. https://doi.org/10.1146/annurev-arplant-050718-100320.
  4. Kim, K.; Lee, S. C.; Lee, J.; Yu, Y.; Yang, K.; Choi, B. S.; Koh, H. J.; Waminal, N. E.; Choi, H. il; Kim, N. H.; et al. J. Complete Chloroplast and Ribosomal Sequences for 30 Accessions Elucidate Evolution of Oryza AA Genome Species. Rep. 2015, 5 (September), 1–13. https://doi.org/10.1038/srep15655.
  5. Nadir, S.; Xiong, H. B.; Zhu, Q.; Zhang, X. L.; Xu, H. Y.; Li, J.; Dongchen, W.; Henry, D.; Guo, X. Q.; Khan, S.; Suh, H. S.; Lee, D. S.; Chen, L. J. Weedy Rice in Sustainable Rice Production. A Review. Sustain. Dev. 2017, 37 (5). https://doi.org/10.1007/s13593-017-0456-4.
  6. Chauhan, B. S.; Jabran, K.; Mahajan, G. Rice Production Worldwide; 2017.Vol. 247. Cham: Springer International Publishing, 2017. https://doi.org/10.1007/978-3-319-47516-5.
  7. Viana, V. E.; Pegoraro, C.; Busanello, C.; Costa de Oliveira, A. Mutagenesis in Rice: The Basis for Breeding a New Super Plant. Plant Sci. 2019, 10 (1326), 1–28. https://doi.org/10.3389/fpls.2019.01326.
  8. Serrat, X.; Esteban, R.; Guibourt, N.; Moysset, L.; Nogués, S.; Lalanne, E. EMS Mutagenesis in Mature Seed-Derived Rice Calli as a New Method for Rapidly Obtaining TILLING Mutant Populations. Plant Methods 2014, 10 (1). https://doi.org/10.1186/1746-4811-10-5.
  9. Soriano, J. D. Mutagenic Effects of Gamma Radiation on Rice. gaz. 1961, 123 (1), 57–63.
  10. Romero, F. M.; Gatica-Arias, A. CRISPR/Cas9: Development and Application in Rice Breeding. Rice Sci. 2019, 26 (5), 265–281. https://doi.org/10.1016/j.rsci.2019.08.001.
  11. Hernández-Soto, A.; Echeverría-Beirute, F.; Abdelnour-Esquivel, A.; Valdez-Melara, M.; Boch, J.; Gatica-Arias, A. Rice Breeding in the New Era: Comparison of Useful Agronomic Traits. Plant Biol. 2021, 27, 100211. https://doi.org/10.1016/j.cpb.2021.100211.
  12. Oladosu, Y.; Rafii, M. Y.; Abdullah, N.; Hussin, G.; Ramli, A.; Rahim, H. A.; Miah, G.; Usman, M. Principle and Application of Plant Mutagenesis in Crop Improvement: A Review. Biotechnol. Equipment 2016, 30 (1), 1–16. https://doi.org/10.1080/13102818.2015.1087333.
  13. Zain, C. R. C. M.; Kadhimi, A. A.; Alhasnawi, A. N.; Isahak, A.; Mohamad, A.; Doni, F.; Yusoff, W. M. W. Enhancing of Drought-Tolerant Rice (Oryza Sativa) Variety MRQ74 through Gamma Radiation and in vitro Biotech. 2016, 15 (6), 125–134. https://doi.org/10.3923/biotech.2016.125.134.
  14. Abdelnour-Esquivel, A.; Perez, J.; Rojas, M.; Vargas, W.; Gatica-Arias, A. Use of Gamma Radiation to Induce Mutations in Rice (Oryza sativa) and the Selection of Lines with Tolerance to Salinity and Drought. In Vitro Cell. Dev. Biol. - Plant 2020, 56 (1), 88–97. https://doi.org/10.1007/s11627-019-10015-5.
  15. Fehér, A. Callus, Dedifferentiation, Totipotency, Somatic Embryogenesis: What These Terms Mean in the Era of Molecular Plant Biology? Plant Sci. 2019, 10 (536). https://doi.org/10.3389/fpls.2019.00536.
  16. Binte Mostafiz, S.; Wagiran, A. Efficient Callus Induction and Regeneration in Selected IndicaAgronomy 20188, 77. https://doi.org/10.3390/agronomy8050077
  17. Vega, R.; Vásquez, N.; Espinoza, A. M.; Gatica, A. M.; Valdez-Melara, M. Histology of Somatic Embryogenesis in Rice (Oryza sativa 5272). Rev. Biol. Trop. 2009, 57 (SUPPL. 1), 141–150. https://doi.org/10.15517/rbt.v57i0.21291.
  18. Radziah C.M.Z., C.; Naji Alhasnawi, A.; A. Kadhimi, A.; Isahak, A.; Mohamad, A.; Farshad Ashraf, M.; Doni, F.; Mohtar Wan Yusoff, W. Development of a Technique for Callus Induction and Plant Regeneration in Oryza sativa Var. MRQ74 and MR269. Advance J. Food Sci. Technol. 2017, 13 (3), 128–137. https://doi.org/10.19026/ajfst.13.4149.
  19. Murashige, T.; Skoog, F. A Revised Medium for Rapid Growth and Bio Assays with Tobacco Tissue Cultures. Plant. 1962, 15 (3), 474–497. https://doi.org/10.1111/j.1399-3054.1962.tb08052.x.
  20. Valdez, M.; Muñoz, M.; Vega, J. R.; Espinoza, A. M. Plant Regeneration of Indica Rice (Oryza sativa) Cultivars from Mature Embryo-Derived Calli. Biol. Trop. 1997, 44–45 (3), 13–21. https://doi.org/10.15517/rbt.v44i3.21827.
  21. Pawar, B.; Kale, P.; Bahurupe, J.; Jadhav, A.; Kale, A.; Pawar, S. Proline and Glutamine Improve in vitro Callus Induction and Subsequent Shooting in Rice. Rice Sci. 2015, 22 (6), 283–289. https://doi.org/10.1016/j.rsci.2015.11.001.
  22. Abiri, R.; Maziah, M.; Shaharuddin, N. A.; Yusof, Z. N. B.; Atabaki, N.; Hanafi, M. M.; Sahebi, M.; Azizi, P.; Kalhori, N.; Valdiani, A. Enhancing Somatic Embryogenesis of Malaysian Rice Cultivar MR219 Using Adjuvant Materials in a High-Efficiency Protocol. J Env. Sci. Techn. 2017, 14 (5), 1091–1108. https://doi.org/10.1007/s13762-016-1221-y.
  23. Kashtwari, M.; Mansoor, S.; Wani, A. A.; Najar, M. A.; Deshmukh, R. K.; Baloch, F. S.; Abidi, I.; Zargar, S. M. Random Mutagenesis in Vegetatively Propagated Crops: Opportunities, Challenges and Genome Editing Prospects. Biol. Rep. 2021. https://doi.org/10.1007/s11033-021-06650-0.
  24. Hussain, Z.; Khan, M. H.; Bano, R.; Rashid, H.; Chaudhry, Z. Protocol Optimization for Efficient Callus Induction and Regeneration in Three Pakistani Rice Cultivars. J. Bot. 2010, 42 (2), 879–887.
  25. Ming, N. J.; Mostafiz, S. B.; Johon, N. S.; Zulkifli, N. S. A.; Wagiran, A. Combination of Plant Growth Regulators, Maltose, and Partial Desiccation Treatment Enhance Somatic Embryogenesis in Selected Malaysian Rice Cultivar. Plants 2019, 8 (6). https://doi.org/10.3390/plants8060144.
  26. Lee, S. T.; Huang, W. L. Cytokinin, Auxin, and Abscisic Acid Affects Sucrose Metabolism Conduce to de Novo Shoot Organogenesis in Rice (Oryza Sativa L.) Callus. Stud. 2013, 54 (1), 1–11. https://doi.org/10.1186/1999-3110-54-5.
  27. Zhang, K.; Su, J.; Xu, M.; Zhou, Z.; Zhu, X.; Ma, X.; Hou, J.; Tan, L.; Zhu, Z.; Cai, H.; Liu, F.; Sun, H.; Gu, P.; Li, C.; Liang, Y.; Zhao, W.; Sun, C.; Fu, Y. A Common Wild Rice-Derived BOC1 Allele Reduces Callus Browning in Indica Rice Transformation. Commun. 2020, 11 (1). https://doi.org/10.1038/s41467-019-14265-0.
  28. Debnath S.; Chandel R.K.; Devi K.; Khan Z. Mechanism and Molecular Response of Induced Genotoxicity and Oxidative Stress in Plants. In: Khan Z., Ansari M.Y.K., Shahwar D. (eds) Induced Genotoxicity and Oxidative Stress in Plants 2021. Springer, Singapore. https://doi.org/10.1007/978-981-16-2074-4_8
  29. Xie, X.; He, Z.; Chen, N.; Tang, Z.; Wang, Q.; Cai, Y. The Roles of Environmental Factors in Regulation of Oxidative Stress in Plant. BioMed Res. Int. 2019, 2019, 21–27. https://doi.org/10.1155/2019/9732325.
  30. IBM Corp. Released 2020. IBM SPSS Statistics for Windows, Version 27.0. Armonk, NY: IBM Corp.
  31. Sudhakar, D.; Duc, L. T.; Bong, B. B.; Tinjuangjun, P.; Maqbool, S. B.; Valdez, M.; Jefferson, R.; Christou, P. An Efficient Rice Transformation System Utilizing Mature Seed-Derived Explants and a Portable, Inexpensive Particle Bombardment Device. Transgenic Res. 1998, 7 (4), 289–294. https://doi.org/10.1023/A:1008870012568.
  32. Schneider, C., Rasband, W. & Eliceiri, K. NIH Image to ImageJ: 25 years of image analysis. Methods 2012, 9, 671–675. https://doi.org/10.1038/nmeth.2089
